# Updates of Equilibrium Prop Match Gradients of Backprop Through Time in an RNN with Static Input

**Maxence Ernoult**[1,2], **Julie Grollier**[2], **Damien Querlioz**[1], **Yoshua Bengio**[3,4], **Benjamin Scellier**[3]

[1]Centre de Nanosciences et de Nanotechnologies, Université Paris Sud, Université Paris-Saclay
[2]Unité Mixte de Physique, CNRS, Thales, Université Paris-Sud, Université Paris-Saclay
[3]Mila, Université de Montréal
[4]Canadian Institute for Advanced Research

## Abstract

Equilibrium Propagation (EP) is a biologically inspired learning algorithm for convergent recurrent neural networks, i.e. RNNs that are fed by a static input $x$ and settle to a steady state. Training convergent RNNs consists in adjusting the weights until the steady state of output neurons coincides with a target $y$. Convergent RNNs can also be trained with the more conventional Backpropagation Through Time (BPTT) algorithm. In its original formulation EP was described in the case of real-time neuronal dynamics, which is computationally costly. In this work, we introduce a discrete-time version of EP with simplified equations and with reduced simulation time, bringing EP closer to practical machine learning tasks. We first prove theoretically, as well as numerically that the neural and weight updates of EP, computed by *forward-time* dynamics, are step-by-step equal to the ones obtained by BPTT, with gradients computed *backward in time*. The equality is strict when the transition function of the dynamics derives from a primitive function and the steady state is maintained long enough. We then show for more standard discrete-time neural network dynamics that the same property is approximately respected and we subsequently demonstrate training with EP with equivalent performance to BPTT. In particular, we define the first convolutional architecture trained with EP achieving $\sim 1\%$ test error on MNIST, which is the lowest error reported with EP. These results can guide the development of deep neural networks trained with EP.

## 1   Introduction

The remarkable development of deep learning over the past years [LeCun et al., 2015] has been fostered by the use of backpropagation [Rumelhart et al., 1985] which stands as the most powerful algorithm to train neural networks. In spite of its success, the backpropagation algorithm is not biologically plausible [Crick, 1989], and its implementation on GPUs is energy-consuming [Editorial, 2018]. Hybrid hardware-software experiments have recently demonstrated how physics and dynamics can be leveraged to achieve learning with energy efficiency [Romera et al., 2018, Ambrogio et al., 2018]. Hence the motivation to invent novel learning algorithms where both inference and learning could fully be achieved out of core physics.

Many biologically inspired learning algorithms have been proposed as alternatives to backpropagation to train neural networks. Contrastive Hebbian learning (CHL) has been successfully used to train recurrent neural networks (RNNs) with static input that converge to a steady state (or 'equilibrium'), such as Boltzmann machines [Ackley et al., 1985] and real-time Hopfield networks [Movellan, 1991]. CHL proceeds in two phases, each phase converging to a steady state, where the learning rule accommodates the difference between the two equilibria. Equilibrium Propagation (EP) [Scellier and Bengio, 2017] also belongs to the family of CHL algorithms to train RNNs with static input.

In the second phase of EP, the prediction error is encoded as an elastic force nudging the system towards a second equilibrium closer to the target. Interestingly, EP also shares similar features with the backpropagation algorithm, and more specifically recurrent backpropagation (RBP) [Almeida, 1987, Pineda, 1987]. It was proved in Scellier and Bengio [2019] that neural computation in the second phase of EP is equivalent to gradient computation in RBP.

Originally, EP was introduced in the context of leaky-integrate neurons [Scellier and Bengio, 2017, 2019]. Computing their dynamics involves long simulation times, hence limiting EP training experiments to small neural networks. In this paper, we propose a discrete-time formulation of EP. This formulation allows demonstrating an equivalence between EP and BPTT in specific conditions, simplifies equations and speeds up training, and extends EP to standard neural networks including convolutional ones. Specifically, the contributions of the present work are the following:

- We introduce a discrete-time formulation of EP (Section 3.1) of which the original real-time formulation can be seen as a particular case (Section 4.2).

- We show a step-by-step equality between the updates of EP and the gradients of BPTT when the dynamics converges to a steady state and the transition function of the RNN derives from a primitive function (Theorem 1, Figure 1). We say that such an RNN has the property of 'gradient-descending updates' (or GDU property).

- We numerically demonstrate the GDU property on a small network, on fully connected layered and convolutional architectures. We show that the GDU property continues to hold approximately for more standard – *prototypical* – neural networks even if these networks do not exactly meet the requirements of Theorem 1.

- We validate our approach with training experiments on different network architectures using discrete-time EP, achieving similar performance than BPTT. We show that the number of iterations in the two phases of discrete-time EP can be reduced by a factor three to five compared to the original real-time EP, without loss of accuracy. This allows us training the first convolutional architecture with EP, reaching $\sim 1\%$ test error on MNIST, which is the lowest test error reported with EP. Our code is available on-line in Pytorch [1].

## 2  Background

This section introduces the notations and basic concepts used throughout the paper.

### 2.1  Convergent RNNs With Static Input

We consider the supervised setting where we want to predict a target $y$ given an input $x$. The model is a dynamical system - such as a recurrent neural network (RNN) - parametrized by $\theta$ and evolving according to the dynamics:

$$s_{t+1} = F\left(x, s_t, \theta\right). \tag{1}$$

We call $F$ the *transition function*. The input of the RNN at each timestep is static, equal to $x$. Assuming convergence of the dynamics before time step $T$, we have $s_T = s_*$ where $s_*$ is such that

$$s_* = F\left(x, s_*, \theta\right). \tag{2}$$

We call $s_*$ the *steady state* (or fixed point, or equilibrium state) of the dynamical system. The number of timesteps $T$ is a hyperparameter chosen large enough to ensure $s_T = s_*$. The goal of learning is to optimize the parameter $\theta$ to minimize the loss:

$$\mathcal{L}^* = \ell\left(s_*, y\right), \tag{3}$$

where the scalar function $\ell$ is called *cost function*. Several algorithms have been proposed to optimize the loss $\mathcal{L}^*$, including Recurrent Backpropagation (RBP) [Almeida, 1987, Pineda, 1987] and Equilibrium Propagation (EP) [Scellier and Bengio, 2017]. Here, we present Backpropagation Through Time (BPTT) and Equilibrium Propagation (EP) and some of the inner mechanisms of these two algorithms, so as to enunciate the main theoretical result of this paper (Theorem 1).

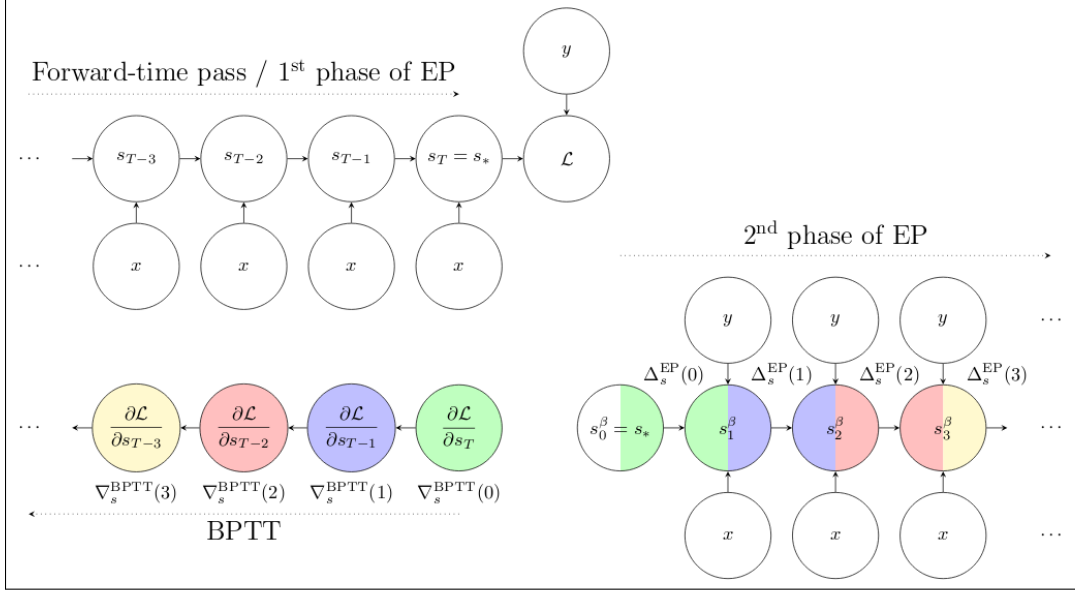

Figure 1: Illustration of the property of Gradient-Descending Updates (GDU property). **Top left.** Forward-time pass (or 'first phase') of an RNN with static input $x$ and target $y$. The final state $s_T$ is the steady state $s_*$. **Bottom left.** Backprop through time (BPTT). **Bottom right.** Second phase of equilibrium prop (EP). The starting state in the second phase is the final state of the first phase, i.e. the steady state $s_*$. **GDU Property (Theorem 1).** Step by step correspondence between the neural updates $\Delta_s^{\mathrm{EP}}(t)$ in the second phase of EP and the gradients $\nabla_s^{\mathrm{BPTT}}(t)$ of BPTT. Corresponding computations in EP and BPTT at timestep $t = 0$ (resp. $t = 1, 2, 3$) are colored in green (resp. blue, red, yellow). Forward-time computation in EP corresponds to backward-time computation in BPTT.

## 2.2 Backpropagation Through Time (BPTT)

With frameworks implementing automatic differentiation, optimization by gradient descent using Backpropagation Through Time (BPTT) has become the standard method to train RNNs. In particular BPTT can be used for a convergent RNN such as the one that we study here. To this end, we consider the loss after $T$ iterations (i.e. the cost of the final state $s_T$), denoted $\mathcal{L} = \ell\left(s_T, y\right)$, and we substitute $\mathcal{L}$ as a proxy [2] for the loss at the steady state $\mathcal{L}^*$. The gradients of $\mathcal{L}$ can be computed with BPTT.

In order to state our Theorem 1 (Section 3.2), we recall some of the inner working mechanisms of BPTT. Eq. (1) can be rewritten in the form $s_{t+1} = F\left(x, s_t, \theta_{t+1} = \theta\right)$, where $\theta_t$ denotes the parameter of the model at time step $t$, the value $\theta$ being shared across all time steps. This way of rewriting Eq. (1) enables us to define the partial derivative $\frac{\partial \mathcal{L}}{\partial \theta_t}$ as the sensitivity of the loss $\mathcal{L}$ with respect to $\theta_t$ when $\theta_1, \ldots \theta_{t-1}, \theta_{t+1}, \ldots \theta_T$ remain fixed (set to the value $\theta$). With these notations, the gradient $\frac{\partial \mathcal{L}}{\partial \theta}$ reads as the sum:

$$\frac{\partial \mathcal{L}}{\partial \theta} = \frac{\partial \mathcal{L}}{\partial \theta_1} + \frac{\partial \mathcal{L}}{\partial \theta_2} + \cdots + \frac{\partial \mathcal{L}}{\partial \theta_T}. \tag{4}$$

BPTT computes the 'full' gradient $\frac{\partial \mathcal{L}}{\partial \theta}$ by computing the partial derivatives $\frac{\partial \mathcal{L}}{\partial s_t}$ and $\frac{\partial \mathcal{L}}{\partial \theta_t}$ iteratively and efficiently, backward in time, using the chain rule of differentiation. Subsequently, we denote the gradients that BPTT computes:

$$\forall t \in [0, T-1] : \begin{cases} \nabla_s^{\mathrm{BPTT}}(t) = \dfrac{\partial \mathcal{L}}{\partial s_{T-t}} \\ \nabla_\theta^{\mathrm{BPTT}}(t) = \dfrac{\partial \mathcal{L}}{\partial \theta_{T-t}}, \end{cases} \tag{5}$$

so that

$$\frac{\partial \mathcal{L}}{\partial \theta} = \sum_{t=0}^{T-1} \nabla_\theta^{\mathrm{BPTT}}(t). \tag{6}$$

More details about BPTT are provided in Appendix A.2.

## 3 Equilibrium Propagation (EP) - Discrete Time Formulation

### 3.1 Algorithm

In its original formulation, Equilibrium Propagation (EP) was introduced in the case of real-time dynamics [Scellier and Bengio, 2017, 2019]. The first theoretical contribution of this paper is to adapt the theory of EP to discrete-time dynamics.[3] EP is an alternative algorithm to compute the gradient of $\mathcal{L}^*$ in the particular case where the transition function $F$ derives from a scalar function $\Phi$, i.e. with $F$ of the form $F(x, s, \theta) = \frac{\partial \Phi}{\partial s}(x, s, \theta)$. In this setting, the dynamics of Eq. (1) rewrites:

$$\forall t \in [0, T-1], \qquad s_{t+1} = \frac{\partial \Phi}{\partial s}(x, s_t, \theta). \tag{7}$$

This constitutes the first phase of EP. At the end of the first phase, we have reached steady state, i.e. $s_T = s_*$. In the second phase of EP, starting from the steady state $s_*$, an extra term $\beta \frac{\partial \ell}{\partial s}$ (where $\beta$ is a positive scaling factor) is introduced in the dynamics of the neurons and acts as an external force nudging the system dynamics towards decreasing the cost function $\ell$. Denoting $s_0^\beta, s_1^\beta, s_2^\beta, \ldots$ the sequence of states in the second phase (which depends on the value of $\beta$), the dynamics is defined as

$$s_0^\beta = s_* \qquad \text{and} \qquad \forall t \geq 0, \quad s_{t+1}^\beta = \frac{\partial \Phi}{\partial s}\left(x, s_t^\beta, \theta\right) - \beta \frac{\partial \ell}{\partial s}\left(s_t^\beta, y\right). \tag{8}$$

The network eventually settles to a new steady state $s_*^\beta$. It was shown in Scellier and Bengio [2017] that the gradient of the loss $\mathcal{L}^*$ can be computed based on the two steady states $s_*$ and $s_*^\beta$. More specifically, [4] in the limit $\beta \to 0$,

$$\frac{1}{\beta}\left(\frac{\partial \Phi}{\partial \theta}\left(x, s_*^\beta, \theta\right) - \frac{\partial \Phi}{\partial \theta}\left(x, s_*, \theta\right)\right) \to -\frac{\partial \mathcal{L}^*}{\partial \theta}. \tag{9}$$

In fact, we can prove a stronger result. For fixed $\beta > 0$ we define the neural and weight updates

$$\forall t \geq 0 : \begin{cases} \Delta_s^{\mathrm{EP}}(\beta, t) = \frac{1}{\beta}\left(s_{t+1}^\beta - s_t^\beta\right), \\ \Delta_\theta^{\mathrm{EP}}(\beta, t) = \frac{1}{\beta}\left(\frac{\partial \Phi}{\partial \theta}\left(x, s_{t+1}^\beta, \theta\right) - \frac{\partial \Phi}{\partial \theta}\left(x, s_t^\beta, \theta\right)\right), \end{cases} \tag{10}$$

and note that Eq. (9) rewrites as the following telescoping sum:

$$\sum_{t=0}^{\infty} \Delta_\theta^{\mathrm{EP}}(\beta, t) \to -\frac{\partial \mathcal{L}^*}{\partial \theta} \qquad \text{as} \qquad \beta \to 0. \tag{11}$$

We can now state our main theoretical result (Theorem 1 below).

### 3.2 Forward-Time Dynamics of EP Compute Backward-Time Gradients of BPTT

BPTT and EP compute the gradient of the loss in very different ways: while the former algorithm iteratively adds up gradients going backward in time, as in Eq. (6), the latter algorithm adds up weight updates going forward in time, as in Eq. (11). In fact, under a condition stated below, the sums are equal term by term: there is a step-by-step correspondence between the two algorithms.

**Theorem 1** (Gradient-Descending Updates, GDU). *Consider the setting with a transition function of the form $F(x, s, \theta) = \frac{\partial \Phi}{\partial s}(x, s, \theta)$. Let $s_0, s_1, \ldots, s_T$ be the convergent sequence of states and denote $s_* = s_T$ the steady state. If we further assume that there exists some step $K$ where $0 < K \leq T$ such that $s_* = s_T = s_{T-1} = \ldots s_{T-K}$, then, in the limit $\beta \to 0$, the first $K$ updates in the second phase of EP are equal to the negatives of the first $K$ gradients of BPTT, i.e.*

$$\forall t = 0, 1, \ldots, K : \begin{cases} \Delta_s^{\mathrm{EP}}(\beta, t) \to -\nabla_s^{\mathrm{BPTT}}(t), \\ \Delta_\theta^{\mathrm{EP}}(\beta, t) \to -\nabla_\theta^{\mathrm{BPTT}}(t). \end{cases} \tag{12}$$

We give here a short outline of the proof of Theorem 1 (A complete proof together with a detailed sketch of the proof are provided in Appendix A). The convergence requirement enables to derive the equations satisfied by the neural and weight updates (Lemma 3). Then, the existence of a primitive function ensures that these equations are equal to those satisfied by the gradients of BPTT (Lemma 2), with same initial conditions, yielding the desired equality (Theorem 1).

Note that other algorithms such as RTRL [Williams and Zipser, 1989] and UORO [Tallec and Ollivier, 2017] also compute the gradients by forward-time dynamics.

## 4   Experiments

This section uses Theorem 1 as a tool to design neural networks that are trainable with EP: if a model satisfies the GDU property of Eq. 12, then we expect EP to perform as well as BPTT on this model. After introducing our protocol (Section 4.1), we define the *energy-based setting* and *prototypical setting* where the conditions of Theorem 1 are exactly and approximately met respectively (Section 4.2). We show the GDU property on a toy model (Fig. 2) and on fully connected layered architectures in the two settings (Section 4.3). We define a convolutional architecture in the prototypical setting (Section 4.4) which also satisfies the GDU property. Finally, we validate our approach by training [5] these models with EP and BPTT (Table 1).

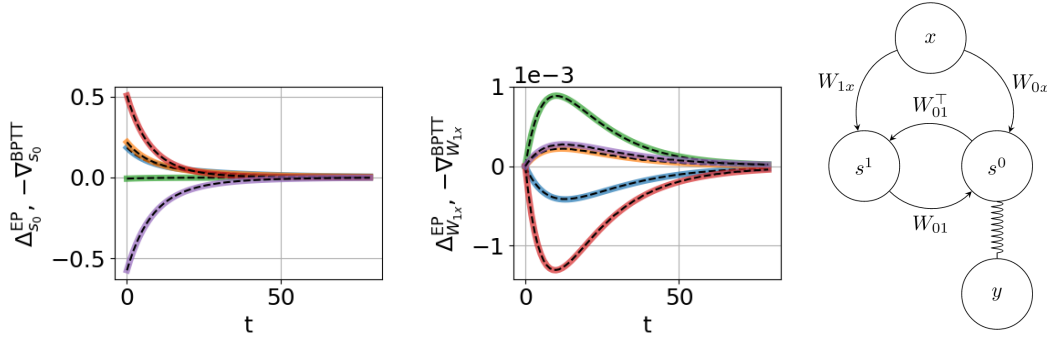

Figure 2: Demonstrating the property of gradient-descending updates in the energy-based setting on a toy model with dummy data $x$ and a target $y$ elastically nudging the output neurons $s^0$ (right). Dashed and solid lines represent $\Delta^{\mathrm{EP}}$ and $-\nabla^{\mathrm{BPTT}}$ processes respectively and perfectly coincide for 5 randomly selected neurons (left) and synapses (middle). Each randomly selected neuron or synapse corresponds to one color. Details can be found in Appendix C.2.

### 4.1   Demonstrating the Property of Gradient-Descending Updates (GDU Property)

**Property of Gradient-Descending Updates.**   We say that a convergent RNN model fed with a fixed input has the *GDU property* if during the second phase, the updates it computes by EP ($\Delta^{\mathrm{EP}}$) on the one hand and the gradients it computes by BPTT ($-\nabla^{\mathrm{BPTT}}$) on the other hand are 'equal' - or 'approximately equal', as measured per the RelMSE (Relative Mean Squared Error) metric.

**Relative Mean Squared Error (RelMSE).**   In order to quantitatively measure how well the GDU property is satisfied, we introduce a metric which we call Relative Mean Squared Error (RelMSE)

such that RelMSE($\Delta^{\mathrm{EP}}$, -$\nabla^{\mathrm{BPTT}}$) measures the distance between $\Delta^{\mathrm{EP}}$ and $-\nabla^{\mathrm{BPTT}}$ processes, averaged over time, over neurons or synapses (layer-wise) and over a mini-batch of samples - see Appendix C.1 for the details.

**Protocol.** In order to measure numerically if a given model satisfies the GDU property, we proceed as follows. Considering an input $x$ and associated target $y$, we perform the first phase for T steps. Then we perform on the one hand BPTT for $K$ steps (to compute the gradients $\nabla^{\mathrm{BPTT}}$), on the other hand EP for $K$ steps (to compute the neural updates $\Delta^{\mathrm{EP}}$) and compare the gradients and neural updates provided by the two algorithms, either qualitatively by looking at the plots of the curves (as in Figs. 2 and 4), or quantitatively by computing their RelMSE (as in Fig. 3).

## 4.2 Energy-Based Setting and Prototypical Setting

**Energy-based setting.** The system is defined in terms of a primitive function of the form:

$$\Phi_\epsilon(s, W) = (1 - \epsilon)\frac{1}{2}\|s\|^2 + \epsilon\,\sigma(s)^\top \cdot W \cdot \sigma(s), \tag{13}$$

where $\epsilon$ is a discretization parameter, $\sigma$ is an activation function and $W$ is a symmetric weight matrix. In this setting, we consider $\Delta^{\mathrm{EP}}(\beta\epsilon, t)$ instead of $\Delta^{\mathrm{EP}}(\beta, t)$ and write $\Delta^{\mathrm{EP}}(t)$ for simplicity, so that:

$$\Delta_s^{\mathrm{EP}}(t) = \frac{s_{t+1}^{\beta\epsilon} - s_t^{\beta\epsilon}}{\beta\epsilon}, \quad \Delta_W^{\mathrm{EP}}(t) = \frac{1}{\beta}\left(\sigma\left(s_{t+1}^{\beta\epsilon}\right)^\top \cdot \sigma\left(s_{t+1}^{\beta\epsilon}\right) - \sigma\left(s_t^{\beta\epsilon}\right)^\top \cdot \sigma\left(s_t^{\beta\epsilon}\right)\right). \tag{14}$$

With $\Phi_\epsilon$ as a primitive function and with the hyperparameter $\beta$ rescaled by a factor $\epsilon$, we recover the discretized version of the real-time setting of Scellier and Bengio [2017], i.e. the Euler scheme of $\frac{ds}{dt} = -\frac{\partial E}{\partial s} - \beta\frac{\partial \ell}{\partial s}$ with $E = \frac{1}{2}\|s\|^2 - \sigma(s)^\top \cdot W \cdot \sigma(s)$ – see Appendix B.2, where the link between discrete-time dynamics and real-time dynamics is explained. Fig. 2 qualitatively demonstrates Theorem 1 in this setting on a toy model.

**Prototypical setting.** In this case, the dynamics of the system does not derive from a primitive function $\Phi$. Instead, the dynamics is directly defined as:

$$s_{t+1} = \sigma\left(W \cdot s_t\right). \tag{15}$$

Again, $W$ is assumed to be a symmetric matrix. The dynamics of Eq. (15) is a standard and simple neural network dynamics. Although the model is not defined in terms of a primitive function, note that $s_{t+1} \approx \frac{\partial \Phi}{\partial s}(s_t, W)$ with $\Phi(s, W) = \frac{1}{2}s^\top \cdot W \cdot s$ if we ignore the activation function $\sigma$. Following Eq. (10), we define:

$$\Delta_s^{\mathrm{EP}}(t) = \frac{1}{\beta}\left(s_{t+1}^\beta - s_t^\beta\right), \qquad \Delta_W^{\mathrm{EP}}(t) = \frac{1}{\beta}\left(s_{t+1}^{\beta\top} \cdot s_{t+1}^\beta - s_t^{\beta\top} \cdot s_t^\beta\right). \tag{16}$$

## 4.3 Effect of Depth and Approximation

We consider a fully connected layered architecture where layers $s^n$ are labelled in a backward fashion: $s^0$ denotes the output layer, $s^1$ the last hidden layer, and so forth. Two consecutive layers are reciprocally connected with tied weights with the convention that $W_{n,n+1}$ connects $s^{n+1}$ to $s^n$. We study this architecture in the energy-based and prototypical setting as described per Equations (13) and (15) respectively, with corresponding weight updates (14) and (16) - see details in Appendix C.3 and C.4. We study the GDU property layer-wise, e.g. RelMSE($\Delta_{s^n}^{\mathrm{EP}}$, -$\nabla_{s^n}^{\mathrm{BPTT}}$) measures the distance between the $\Delta_{s^n}^{\mathrm{EP}}$ and $-\nabla_{s^n}^{\mathrm{BPTT}}$ processes, averaged over all elements of layer $s^n$.

We display in Fig. 3 the RelMSE, layer-wise for one, two and three hidden layered architecture (from left to right), in the energy-based (upper panels) and prototypical (lower panels) settings, so that each architecture in a given setting is displayed in one panel - see Appendix C.3 and C.4 for a detailed description of the hyperparameters and curve samples. In terms of RelMSE, we can see that the GDU property is best satisfied in the energy-based setting with one hidden layer where RelMSE is around $\sim 10^{-2}$ (top left). When adding more hidden layers in the energy-based setting (top middle and top right), the RelMSE increases to $\sim 10^{-1}$, with a greater RelMSE when going away from the output layer. The same is observed in the prototypical setting when we add more hidden layers (lower

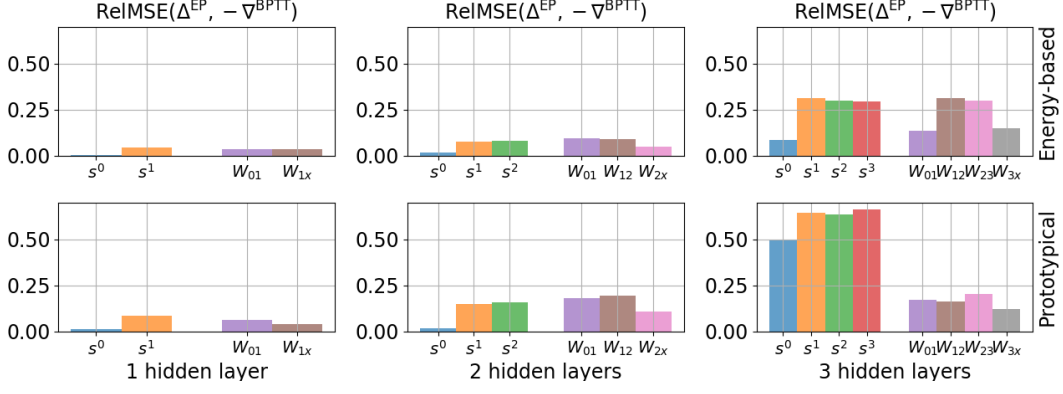

Figure 3: RelMSE analysis in the energy-based (top) and prototypical (bottom) setting. For one given architecture, each bar is labelled by a layer or synapses connecting two layers, e.g. the orange bar above $s^1$ represents $\text{RelMSE}(\Delta_{s^1}^{\text{EP}}, -\nabla_{s^1}^{\text{BPTT}})$. For each architecture, the recurrent hyperparameters $T$, $K$ and $\beta$ have been tuned to make the $\Delta^{\text{EP}}$ and $-\nabla^{\text{BPTT}}$ processes match best.

panels). Compared to the energy-based setting, although the RelMSEs associated with neurons are significantly higher in the prototypical setting, the RelMSEs associated with synapses are similar or lower. On average, the weight updates provided by EP match well the gradients of BPTT, in the energy-based setting as well as in the prototypical setting.

## 4.4 Convolutional Architecture

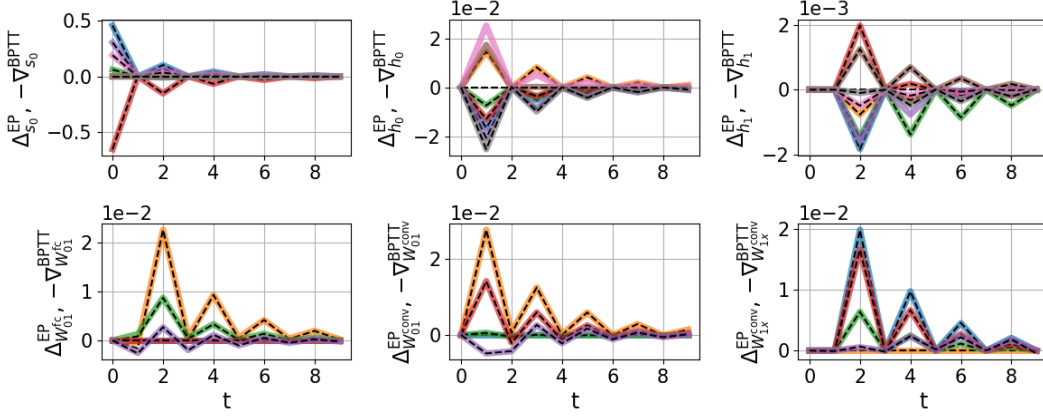

Figure 4: Demonstrating the GDU property with the convolutional architecture on MNIST. Dashed and continuous lines represent $\Delta^{\text{EP}}$ and $-\nabla^{\text{BPTT}}$ processes respectively, for 5 randomly selected neurons (top) and synapses (bottom) in each layer. Each randomly selected neuron or synapse corresponds to one color. Dashed and continuous lines mostly coincide. Some $\Delta^{\text{EP}}$ processes collapse to zero as an effect of the non-linearity, see Appendix D for details. Interestingly, the $\Delta_s^{\text{EP}}$ and $-\nabla_s^{\text{BPTT}}$ processes are saw-teeth-shaped ; Appendix C.6 accounts for this phenomenon.

In our convolutional architecture, $h^n$ and $s^n$ denote convolutional and fully connected layers respectively. $W_{n,n+1}^{\text{fc}}$ and $W_{n,n+1}^{\text{conv}}$ denote the fully connected weights connecting $s^{n+1}$ to $s^n$ and the filters connecting $h^{n+1}$ to $h^n$, respectively. We define the dynamics as:

$$\begin{cases} s_{t+1}^n &= \sigma\left(W_{nn+1}^{\text{fc}} \cdot s_t^{n+1} + W_{n-1n}^{\text{fc}^\top} \cdot s_t^{n-1}\right) \\ h_{t+1}^n &= \sigma\left(\mathcal{P}\left(W_{n,n+1}^{\text{conv}} * h_t^{n+1}\right) + \tilde{W}_{n-1,n}^{\text{conv}} * \mathcal{P}^{-1}\left(h_t^{n-1}\right)\right), \end{cases} \quad (17)$$

where $*$ and $\mathcal{P}$ denote convolution and pooling, respectively. Transpose convolution is defined through the convolution by the flipped kernel $\tilde{W}^{\text{conv}}$ and $\mathcal{P}^{-1}$ denotes inverse pooling - see Appendix D for a

Table 1: Training results on MNIST with EP benchmarked against BPTT, in the energy-based and prototypical settings. "EB" and "P" respectively denote "energy-based" and "prototypical", "-#h" stands for the number of hidden layers and WCT for "Wall-clock time" in $hours : minutes$. We indicate over five trials the mean and standard deviation for the test error, the mean error in parenthesis for the train error. $T$ (resp. $K$) is the number of iterations in the first (resp. second) phase.

| | EP (error %) | | BPTT (error %) | | T | K | Epochs | WCT |
|---|---|---|---|---|---|---|---|---|
| | Test | Train | Test | Train | | | | |
| EB-1h | $2.06 \pm 0.17$ | $(0.13)$ | $2.11 \pm 0.09$ | $(0.46)$ | 100 | 12 | 30 | $1 : 33$ |
| EB-2h | $2.01 \pm 0.21$ | $(0.11)$ | $2.02 \pm 0.12$ | $(0.29)$ | 500 | 40 | 50 | $16 : 04$ |
| P-1h | $2.00 \pm 0.13$ | $(0.20)$ | $2.00 \pm 0.12$ | $(0.55)$ | 30 | 10 | 30 | $\mathbf{0 : 17}$ |
| P-2h | $1.95 \pm 0.10$ | $(0.14)$ | $2.09 \pm 0.12$ | $(0.37)$ | 100 | 20 | 50 | $\mathbf{1 : 56}$ |
| P-3h | $2.01 \pm 0.18$ | $(0.10)$ | $2.30 \pm 0.17$ | $(0.32)$ | 180 | 20 | 100 | $\mathbf{8 : 27}$ |
| P-conv | $\mathbf{1.02 \pm 0.04}$ | $(0.54)$ | $0.88 \pm 0.06$ | $(0.12)$ | 200 | 10 | 40 | $\mathbf{8 : 58}$ |

Table 2: Best training results on MNIST with EP reported in the literature.

| | EP (error %) | |
|---|---|---|
| | Test | Train |
| [Scellier and Bengio, 2017] | $\sim 2.2$ | $(\sim 0)$ |
| [O'Connor et al., 2018] | 2.37 | $(0.15)$ |
| [O'Connor et al., 2019] | 2.19 | |

precise definition of these operations and their inverse. Noting $N_{\text{fc}}$ and $N_{\text{conv}}$ the number of fully connected and convolutional layers respectively, we can define the function:

$$\Phi(x, \{s^n\}, \{h^n\}) = \sum_{n=0}^{N_{\text{conv}}-1} h^n \bullet \mathcal{P}\left(W_{n,n+1}^{\text{conv}} * h^{n+1}\right) + \sum_{n=0}^{N_{\text{fc}}-1} s^{n\top} \cdot W_{n,n+1}^{\text{fc}} \cdot s^{n+1}, \quad (18)$$

with $\bullet$ denoting generalized scalar product. We note that $s_{t+1}^n \approx \frac{\partial \Phi}{\partial s}(t)$ and $h_{t+1}^n \approx \frac{\partial \Phi}{\partial h}(t)$ if we ignore the activation function $\sigma$. We define $\Delta_{s^n}^{\text{EP}}$, $\Delta_{h^n}^{\text{EP}}$ and $\Delta_{W^{\text{fc}}}^{\text{EP}}$ as in Eq. 16. As for $\Delta_{W^{\text{conv}}}^{\text{EP}}$, we follow the definition of Eq. (10):

$$\Delta_{W_{nn+1}^{\text{conv}}}^{\text{EP}}(t) = \frac{1}{\beta}\left(\mathcal{P}^{-1}(h_{t+1}^{n,\beta}) * h_{t+1}^{n+1,\beta} - \mathcal{P}^{-1}(h_t^{n,\beta}) * h_t^{n+1,\beta}\right) \quad (19)$$

As can be seen in Fig. 4, the GDU property is qualitatively very well satisfied: Eq. (19) can thus be safely used as a learning rule. More precisely however, some $\Delta_{s^n}^{\text{EP}}$ and $\Delta_{h^n}^{\text{EP}}$ processes collapse to zero as an effect of the non-linearity used (see Appendix C for greater details): the EP error signals cannot be transmitted through saturated neurons, resulting in a RelMSE of $\sim 10^{-1}$ for the network parameters - see Fig. 15 in Appendix D.

## 5  Discussion

Table 1 shows the accuracy results on MNIST of several variations of our approach and of the literature. First, EP overall performs as well or practically as well as BPTT in terms of test accuracy in all situations. Second, no degradation of accuracy is seen between using the prototypical (P) rather than the energy-based (EB) setting, although the prototypical setting requires three to five times less time steps in the first phase (T) and cuts the simulation time by a factor five to eight. Finally, the best EP result, $\sim 1\%$ test error, is obtained with our convolutional architecture. This is also the best performance reported in the literature on MNIST training with EP. BPTT achieves $0.90\%$ test error using the same architecture. This slight degradation is due to saturated neurons which do no route error signals (as reported in the previous section). The prototypical situation allows using highly reduced number of time steps in the first phase than Scellier and Bengio [2017] and O'Connor et al. [2018]. On the other hand, O'Connor et al. [2019] manages to cut this number even more. This comes at the cost of using an extra network to learn proper initial states for the EP network, which is not needed in our approach.

Overall, our work broadens the scope of EP from its original formulation for biologically motivated real-time dynamics and sheds new light on its practical understanding. We first extended EP to a discrete-time setting, which reduces its computational cost and allows addressing situations closer to conventional machine learning. [6] Theorem 1 demonstrated that the gradients provided by EP are strictly equal to the gradients computed with BPTT in specific conditions. Our numerical experiments confirmed the theorem and showed that its range of applicability extends well beyond the original formulation of EP to prototypical neural networks widely used today. These results highlight that, in principle, EP can reach the same performance as BPTT on benchmark tasks, for RNN models with *fixed input*. One limitation of our theory however is that it has yet to be adapted to sequential data: such an extension would require to capture and learn correlations between successive equilibrium states corresponding to different inputs.

Layer-wise analysis of the gradients computed by EP and BPTT show that the deeper the layer, the more difficult it becomes to ensure the GDU property. On top of non-linearity effects, this is mainly due to the fact that the deeper the network, the longer it takes to reach equilibrium.

While this may be a conundrum for current processors, it should not be an issue for alternative computing schemes. Physics research is now looking at neuromorphic computing approaches that leverage the transient dynamics of physical devices for computation [Torrejon et al., 2017, Romera et al., 2018, Feldmann et al., 2019]. In such systems, based on magnetism or optics, dynamical equations are solved directly by the physical circuits and components, in parallel and at speed much higher than processors. On the other hand, in such systems, the nonlocality of backprop is a major concern [Ambrogio et al., 2018]. In this context, EP appears as a powerful approach as computing gradients only requires measuring the system at the end of each phase, and going backward in time is not needed. In a longer term, interfacing the algorithmics of EP with device physics could help cutting drastically the cost of inference and learning of conventional computers, and thereby address one of the biggest technological limitations of deep learning.

## Acknowledgments

The authors would like to thank Joao Sacramento for feedback and discussions, as well as NSERC, CIFAR, Samsung and Canada Research Chairs for funding. Julie Grollier and Damien Querlioz acknowledge funding from the European Research Council, respectively under grants bioSPINspired (682955) and NANOINFER (715872).

## Footnotes

[1] https://github.com/ernoult/updatesEPgradientsBPTT

[2]The difference between the loss $\mathcal{L}$ and the loss $\mathcal{L}^*$ is explained in Appendix B.1.

[3]We explain in Appendix B.2 the relationship between the discrete-time setting (resp. the primitive function $\Phi$) of this paper and the real-time setting (resp. the energy function $E$) of Scellier and Bengio [2017, 2019].

[4]The EP learning rule is a form of contrastive Hebbian learning similar to that of Boltzmann machines [Ackley et al., 1985] and similar to the one presented in Movellan [1991].

[5]For training, the effective weight update is performed at the end of the second phase.

[6]We also expect that our discrete-time formulation would accelerate simulations in the setting of Scellier et al. [2018] where the weights are not tied.

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
