[Supplementary Material]

# Appendix

## A Proof of Theorem 1 - Step-by-Step Equivalence of EP and BPTT

In this section, we prove Theorem 1, which we recall here.

**Theorem 1** (Gradient-Descending Updates, GDU). *Consider the setting with a transition function of the form $F(x, s, \theta) = \frac{\partial \Phi}{\partial s}(x, s, \theta)$. Let $s_0, s_1, \ldots, s_T$ be the convergent sequence of states and denote $s_* = s_T$ the steady state. If we further assume that there exists some step $K$ where $0 < K \leq T$ such that $s_* = s_T = s_{T-1} = \ldots s_{T-K}$, then, in the limit $\beta \to 0$, the first $K$ updates in the second phase of EP are equal to the negatives of the first $K$ gradients of BPTT, i.e.*

$$\forall t = 0, 1, \ldots, K : \begin{cases} \Delta_s^{\mathrm{EP}}(\beta, t) \to -\nabla_s^{\mathrm{BPTT}}(t), \\ \Delta_\theta^{\mathrm{EP}}(\beta, t) \to -\nabla_\theta^{\mathrm{BPTT}}(t). \end{cases} \tag{12}$$

In this section we choose a slightly different convention for the definition of the $\nabla_\theta^{\mathrm{BPTT}}(t)$ and $\Delta_\theta^{\mathrm{EP}}(t)$ processes, with an index shift. We explain in Appendix B.3 why this convention is in fact more natural.

### A.1 Sketch of the Proof

Theorem 1 is a consequence of Lemma 2 and Lemma 3 below. Lemma 2 is itself a straightforward consequence of the formula for back-propagating the gradients in an RNN (Proposition 4 in subsection A.2). As for Lemma 3, it is an intermediary result for the more general version of EP with arbitrary transition function $F$ (presented in subsection A.3).

**Lemma 2.** *In our specific setting with static input $x$, suppose that the network has reached the steady state $s_*$ after $T - K$ steps, i.e.*

$$s_{T-K} = s_{T-K+1} = \cdots = s_{T-1} = s_T = s_*. \tag{20}$$

*Then the first $K$ gradients of BPTT satisfy the recurrence relationship* [7]

$$\nabla_s^{\mathrm{BPTT}}(0) = \frac{\partial \ell}{\partial s}(s_*, y), \tag{21}$$

$$\forall t = 1, 2, \ldots, K, \qquad \nabla_s^{\mathrm{BPTT}}(t) = \frac{\partial F}{\partial s}(x, s_*, \theta)^\top \cdot \nabla_s^{\mathrm{BPTT}}(t-1), \tag{22}$$

$$\forall t = 1, 2, \ldots, K, \qquad \nabla_\theta^{\mathrm{BPTT}}(t) = \frac{\partial F}{\partial \theta}(x, s_*, \theta)^\top \cdot \nabla_s^{\mathrm{BPTT}}(t-1). \tag{23}$$

**Lemma 3.** *Let $\Delta_s^{\mathrm{EP}}(t) = \lim_{\beta \to 0} \Delta_s^{\mathrm{EP}}(\beta, t)$ and $\Delta_\theta^{\mathrm{EP}}(t) = \lim_{\beta \to 0} \Delta_\theta^{\mathrm{EP}}(\beta, t)$ be the neural and weight updates of EP in the limit $\beta \to 0$. They satisfy the recurrence relationship*

$$\Delta_s^{\mathrm{EP}}(0) = -\frac{\partial \ell}{\partial s}(s_*, y), \tag{24}$$

$$\forall t \geq 0, \qquad \Delta_s^{\mathrm{EP}}(t+1) = \frac{\partial F}{\partial s}(x, s_*, \theta) \cdot \Delta_s^{\mathrm{EP}}(t), \tag{25}$$

$$\forall t \geq 0, \qquad \Delta_\theta^{\mathrm{EP}}(t+1) = \frac{\partial F}{\partial \theta}(x, s_*, \theta)^\top \cdot \Delta_s^{\mathrm{EP}}(t). \tag{26}$$

Note that the multiplicative matrix in Eq. 25 is the square matrix $\frac{\partial F}{\partial s}(x, s_*, \theta)$ whereas the one in Eq. 22 is its transpose $\frac{\partial F}{\partial s}(x, s_*, \theta)^\top$. Because of that, the updates $\Delta_s^{\mathrm{EP}}(t)$ and $\Delta_\theta^{\mathrm{EP}}(t)$ of EP on the one hand, and the gradients $\nabla_s^{\mathrm{BPTT}}(t)$ and $\nabla_\theta^{\mathrm{BPTT}}(t)$ of BPTT on the other hand, satisfy different recurrence relationships in general. Except when $F$ is of the form $F(x, s, \theta) = \frac{\partial \Phi}{\partial s}(x, s, \theta)$ ; in this case the Jacobian matrix of the transition function $F$ is the Hessian of $\Phi$, thus is symmetric:

$$\frac{\partial F}{\partial s}(x, s, \theta)^\top = \frac{\partial^2 \Phi}{\partial s^2}(x, s, \theta) = \frac{\partial F}{\partial s}(x, s, \theta). \tag{27}$$

## A.2 Backpropagation Through Time (BPTT)

Recall that we are considering an RNN (with fixed input $x$ and target $y$) whose dynamics $s_0, s_1, \ldots, s_T$ and loss $\mathcal{L}$ are defined by[8]

$$\forall t = 0, 1, \ldots T - 1, \qquad s_{t+1} = F\left(x, s_t, \theta_t = \theta\right), \qquad \mathcal{L} = \ell\left(s_T, y\right). \tag{28}$$

We denote the gradients computed by BPTT

$$\forall t = 0, 1, \ldots T, \qquad \nabla_s^{\mathrm{BPTT}}(t) = \frac{\partial \mathcal{L}}{\partial s_{T-t}}, \tag{29}$$

$$\forall t = 1, 2, \ldots T, \qquad \nabla_\theta^{\mathrm{BPTT}}(t) = \frac{\partial \mathcal{L}}{\partial \theta_{T-t}}. \tag{30}$$

The gradients $\nabla_s^{\mathrm{BPTT}}(t)$ and $\nabla_\theta^{\mathrm{BPTT}}(t)$ are the 'elementary gradients' (as illustrated in Fig. 5) computed as intermediary steps in BPTT in order to compute the 'full gradient' $\frac{\partial \mathcal{L}}{\partial \theta}$.

**Proposition 4** (Backpropagation Through Time). *The gradients $\nabla_s^{\mathrm{BPTT}}(t)$ and $\nabla_\theta^{\mathrm{BPTT}}(t)$ can be computed using the recurrence relationship*

$$\nabla_s^{\mathrm{BPTT}}(0) = \frac{\partial \ell}{\partial s}\left(s_T, y\right), \tag{31}$$

$$\forall t = 1, 2, \ldots, T, \qquad \nabla_s^{\mathrm{BPTT}}(t) = \frac{\partial F}{\partial s}\left(x, s_{T-t}, \theta\right)^\top \cdot \nabla_s^{\mathrm{BPTT}}(t-1), \tag{32}$$

$$\forall t = 1, 2, \ldots, T, \qquad \nabla_\theta^{\mathrm{BPTT}}(t) = \frac{\partial F}{\partial \theta}\left(x, s_{T-t}, \theta\right)^\top \cdot \nabla_s^{\mathrm{BPTT}}(t-1). \tag{33}$$

*Proof of Proposition 4.* This is a direct application of the chain rule of differentiation, using the fact that $s_{t+1} = F\left(x, s_t, \theta\right)$ $\qquad\qquad\square$

## A.3 Equilibrium Propagation (EP) – A Formulation with Arbitrary Transition Function $F$

In this section, we show (Lemma 3 below) that the neural updates $\Delta_s^{\mathrm{EP}}(t)$ and weight updates $\Delta_\theta^{\mathrm{EP}}(t)$ of EP satisfy a recurrence relation similar to the one for the gradients of BPTT (Proposition 4, or more specifically Lemma 2).

In section 3 we have presented EP in the setting where the transition function $F$ derives from a scalar function $\Phi$, i.e. with $F$ of the form $F(x, s, \theta) = \frac{\partial \Phi}{\partial s}(x, s, \theta)$. This hypothesis is necessary to show equality of the updates of EP and the gradients of BPTT (Theorem 1). To better emphasize where this hypothesis is used, we first show an intermediary result (Lemma 3 below) which holds for arbitrary transition function $F$.

First we formulate EP for arbitrary transition function $F$, inspired by the ideas of Scellier et al. [2018]. Recall that at the beginning of the second phase of EP the state of the network is the steady state $s_0^\beta = s_*$ characterized by

$$s_* = F\left(x, s_*, \theta\right), \tag{34}$$

and that, given some value $\beta > 0$ of the hyperparameter $\beta$, the successive neural states $s_1^\beta, s_2^\beta, \ldots$ are defined and computed as follows:

$$\forall t \geq 0, \qquad s_{t+1}^\beta = F\left(x, s_t^\beta, \theta\right) - \beta \frac{\partial \ell}{\partial s}\left(s_t^\beta, y\right). \tag{35}$$

In this more general setting, we redefine the 'neural updates' and 'weight updates' as follows[9]:

$$\forall t \geq 0, \qquad \Delta_s^{\mathrm{EP}}(\beta, t) = \frac{1}{\beta}\left(s_{t+1}^\beta - s_t^\beta\right), \tag{36}$$

$$\forall t \geq 1, \qquad \Delta_\theta^{\mathrm{EP}}(\beta, t) = \frac{1}{\beta}\frac{\partial F}{\partial \theta}\left(x, s_{t-1}^\beta, \theta\right)^\top \cdot \left(s_t^\beta - s_{t-1}^\beta\right). \tag{37}$$

Figure 5: **Top.** Computational graph of an RNN with fixed input $x$ and target $y$, unfolded in time. As usual for RNNs, the parameters $\theta_0, \theta_1, \ldots, \theta_{T-1}$ at each time step share the same value $\theta$. The terminal state of the network is the steady state, i.e. $s_T = s_*$. **Bottom.** Backpropagation Through Time (BPTT) computes the gradients $\frac{\partial \mathcal{L}}{\partial s_T}, \frac{\partial \mathcal{L}}{\partial s_{T-1}}, \ldots, \frac{\partial \mathcal{L}}{\partial s_1}$ and $\frac{\partial \mathcal{L}}{\partial \theta_{T-1}}, \frac{\partial \mathcal{L}}{\partial \theta_{T-2}}, \ldots, \frac{\partial \mathcal{L}}{\partial \theta_0}$ as intermediary steps in order to compute the total gradient $\frac{\partial \mathcal{L}}{\partial \theta}$ as in Eq. 4.

In contrast, recall that in the gradient-based setting of section 3 we had defined

$$\Delta_\theta^{\mathrm{EP}}(\beta, t) = \frac{1}{\beta} \left( \frac{\partial \Phi}{\partial \theta} \left( x, s_t^\beta, \theta \right) - \frac{\partial \Phi}{\partial \theta} \left( x, s_{t-1}^\beta, \theta \right) \right). \tag{38}$$

When $F = \frac{\partial \Phi}{\partial s}$, the definitions of Eq. 37 and Eq. 38 are slightly different, but what matters is that both definitions coincide in the limit $\beta \to 0$. Now that we have redefined $\Delta_s^{\mathrm{EP}}(\beta, t)$ and $\Delta_\theta^{\mathrm{EP}}(\beta, t)$ for general transition function $F$, we can recall our intermediary result:

**Lemma 3.** *Let $\Delta_s^{\mathrm{EP}}(t) = \lim_{\beta \to 0} \Delta_s^{\mathrm{EP}}(\beta, t)$ and $\Delta_\theta^{\mathrm{EP}}(t) = \lim_{\beta \to 0} \Delta_\theta^{\mathrm{EP}}(\beta, t)$ be the neural and weight updates of EP in the limit $\beta \to 0$. They satisfy the recurrence relationship*

$$\Delta_s^{\mathrm{EP}}(0) = -\frac{\partial \ell}{\partial s} \left( s_*, y \right), \tag{24}$$

$$\forall t \geq 0, \qquad \Delta_s^{\mathrm{EP}}(t+1) = \frac{\partial F}{\partial s} \left( x, s_*, \theta \right) \cdot \Delta_s^{\mathrm{EP}}(t), \tag{25}$$

$$\forall t \geq 0, \qquad \Delta_\theta^{\mathrm{EP}}(t+1) = \frac{\partial F}{\partial \theta} \left( x, s_*, \theta \right)^\top \cdot \Delta_s^{\mathrm{EP}}(t). \tag{26}$$

*Proof of Lemma 3.* First of all, in the limit $\beta \to 0$, the weight update $\Delta_\theta^{\mathrm{EP}}(\beta, t)$ of Eq. 37 rewrites

$$\Delta_\theta^{\mathrm{EP}}(t) = \frac{\partial F}{\partial \theta} \left( x, s_*, \theta \right)^\top \cdot \Delta_s^{\mathrm{EP}}(t). \tag{39}$$

Hence Eq. 26. Now we prove Eq. 24-25. Note that the neural update $\Delta_s^{\mathrm{EP}}(\beta, t)$ of Eq. 36 rewrites

$$\Delta_s^{\mathrm{EP}}(t) = \left. \frac{\partial s_{t+1}^\beta}{\partial \beta} \right|_{\beta=0} - \left. \frac{\partial s_t^\beta}{\partial \beta} \right|_{\beta=0}. \tag{40}$$

This is because for every $t \geq 0$ we have $s_t^\beta \to s_*$ as $\beta \to 0$ : starting from $s_0^0 = s_*$, if you set $\beta = 0$ in Eq. 35, then $s_1^0 = s_2^0 = \ldots = s_*$.

Differentiating Eq. 35 with respect to $\beta$, we get

$$\forall t \geq 0, \qquad \frac{\partial s_{t+1}^\beta}{\partial \beta} = \frac{\partial F}{\partial s}\left(x, s_t^\beta, \theta\right) \cdot \frac{\partial s_t^\beta}{\partial \beta} - \frac{\partial \ell}{\partial s}\left(s_t^\beta, y\right) - \beta \frac{\partial^2 \ell}{\partial s^2}\left(s_t^\beta, y\right) \cdot \frac{\partial s_t^\beta}{\partial \beta}. \qquad (41)$$

Letting $\beta \to 0$, we have $s_t^\beta \to s_*$, so that

$$\forall t \geq 0, \qquad \left. \frac{\partial s_{t+1}^\beta}{\partial \beta}\right|_{\beta=0} = \frac{\partial F}{\partial s}\left(x, s_*, \theta\right) \cdot \left.\frac{\partial s_t^\beta}{\partial \beta}\right|_{\beta=0} - \frac{\partial \ell}{\partial s}\left(s_*, y\right). \qquad (42)$$

Since at time $t = 0$ the initial state of the network $s_0^\beta = s_*$ is independent of $\beta$, we have

$$\frac{\partial s_0^\beta}{\partial \beta} = 0. \qquad (43)$$

Using Eq. 42 for $t = 0$ and Eq. 43, we get the initial condition (Eq. 24)

$$\Delta_s^{\mathrm{EP}}(0) = \left.\frac{\partial s_1^\beta}{\partial \beta}\right|_{\beta=0} - \left.\frac{\partial s_0^\beta}{\partial \beta}\right|_{\beta=0} = -\frac{\partial \ell}{\partial s}\left(s_*, y\right). \qquad (44)$$

Moreover, if we take Eq. 42 and subtract itself from it at time step $t - 1$, we get

$$\Delta_s^{\mathrm{EP}}(t + 1) = \frac{\partial F}{\partial s}\left(x, s_*, \theta\right) \cdot \Delta_s^{\mathrm{EP}}(t). \qquad (45)$$

Hence Eq. 25. Hence the result. $\qquad\square$

# B Notations

In this Appendix we motivate some of the distinctions that we make and the notations that we adopt. This includes:

1. the distinction between the loss $\mathcal{L}^* = \ell\left(s_*, y\right)$ and the loss $\mathcal{L} = \ell\left(s_T, y\right)$, which could seem unnecessary since $T$ is chosen such that $s_T = s^*$,
2. the distinction between the 'primitive function' $\Phi(x, s, \theta)$ introduced in this paper and the 'energy function' $E(x, s, \theta)$ used in Scellier and Bengio [2017, 2019],
3. the index shift in the definition of the processes $\nabla_\theta^{\mathrm{BPTT}}(t)$ and $\Delta_\theta^{\mathrm{EP}}(t)$.

## B.1 Difference between $\mathcal{L}^*$ and $\mathcal{L}$

There is a difference between the loss at the steady state $\mathcal{L}^*$ and the loss after $T$ iterations $\mathcal{L}$. To see why the functions $\mathcal{L}^*$ and $\mathcal{L}$ (as functions of $\theta$) are different, we have to come back to the definitions of $s_*$ and $s_T$. Recall that

- $\mathcal{L}^* = \ell\left(s_*, y\right)$ where $s_*$ is the steady state, i.e. characterized by $s_* = F\left(x, s_*, \theta\right)$,
- $\mathcal{L} = \ell\left(s_T, y\right)$ where $s_T$ is the state of the network after $T$ time steps, following the dynamics $s_0 = 0$ and $s_{t+1} = F\left(x, s_t, \theta\right)$.

For the current value of the parameter $\theta$, the hyperparameter $T$ is chosen such that $s_T = s_*$, i.e. such that the network reaches steady state after $T$ time steps. Thus, for this value of $\theta$ we have numerical equality $\mathcal{L}(\theta) = \mathcal{L}^*(\theta)$. However, two functions that have the same value at a given point are not necessarily equal. Similarly, two functions that have the same value at a given point don't necessarily have the same gradient at that point. Here we are in the situation where

1. the functions $\mathcal{L}$ and $\mathcal{L}^*$ (as functions of $\theta$) have the same value at the current value of $\theta$, i.e. $\mathcal{L}(\theta) = \mathcal{L}^*(\theta)$ numerically,
2. the functions $\mathcal{L}$ and $\mathcal{L}^*$ (as functions of $\theta$) are analytically different, i.e. $\mathcal{L} \neq \mathcal{L}^*$.

Since the functions $\mathcal{L}$ and $\mathcal{L}^*$ (as functions of $\theta$) are different, the gradients $\frac{\partial \mathcal{L}^*}{\partial \theta}$ and $\frac{\partial \mathcal{L}}{\partial \theta}$ are also different in general.

## B.2 Difference between the Primitive Function $\Phi$ and the Energy Function $E$

Previous work on EP [Scellier and Bengio, 2017, 2019] has studied real-time dynamics of the form:

$$\frac{ds_t}{dt} = -\frac{\partial E}{\partial s}(x, s_t, \theta).$$
(46)

In contrast, in this paper we study discrete-time dynamics of the form

$$s_{t+1} = \frac{\partial \Phi}{\partial s}(x, s_t, \theta).$$
(47)

Here we explain why we changed the sign convention in the dynamics and why we called $\Phi$ a 'primitive function' rather than an 'energy function'.

While it is useful to think of the primitive function $\Phi$ in the discrete-time setting as an equivalent of the energy function $E$ in the real-time setting, there is an important difference between $E$ and $\Phi$. We argue next that, rather than an energy function, $\Phi$ is much better thought of as a primitive of the transition function $F$. First we show how the two settings are related.

**Casting real-time dynamics to discrete-time dynamics.**  The real-time dynamics of Eq. (46) can be cast to the discrete-time setting of Eq. (47) as follows. The Euler scheme of Eq. (46) with discretization step $\epsilon$ reads:

$$s_{t+1} = s_t - \epsilon \frac{\partial E}{\partial s}(x, s_t, \theta).$$
(48)

This equation rewrites

$$s_{t+1} = \frac{\partial \Phi_\epsilon}{\partial s}(x, s_t, \theta), \qquad \text{where} \qquad \Phi_\epsilon(x, s, \theta) = \frac{1}{2}\|s\|^2 - \epsilon\, E(x, s, \theta).$$
(49)

However, although the real-time dynamics can be mapped to the discrete-time setting, the discrete-time setting is more general. The primitive function $\Phi$ cannot be interpreted in terms of an energy in general, as we show next.

**Why not keep the notation $E$ and the name of 'energy function' in the discrete-time framework?**  In the real-time setting, $s_t$ *follows* the gradient of $E$, so that $E(s_t)$ decreases as time progresses until $s_t$ settles to a (local) minimum of $E$. This property motivates the name of 'energy function' for $E$ by analogy with physical systems whose dynamics settle down to low-energy configurations. In contrast, in the discrete-time setting, $s_t$ *is mapped* onto the gradient of $\Phi$ (at the point $s_t$). In general, there is no guarantee that the discrete-time dynamics of Eq. (47) optimizes $\Phi$ and there is no guarantee that the dynamics of $s_t$ converges to an optimum of $\Phi$. For this reason, there is no reason to call $\Phi$ an 'energy function', since the intuition of optimizing an energy does not hold.

**The scalar function $\Phi$ is better thought of as a primitive function of $F$.**  The name of 'primitive function' for $\Phi$ is motivated by the fact that $\Phi$ is a primitive of the transition function $F$, whose property better captures the assumptions under which the theory of EP holds. To see this, we first rewrite Eq. (47) in the form

$$s_{t+1} = F(x, s_t, \theta),$$
(50)

where $F$ is a transition function (in the state space) of the form

$$F(x, s, \theta) = \frac{\partial \Phi}{\partial s}(x, s, \theta),$$
(51)

with $\Phi(x, s, \theta)$ a scalar function. For the theory of EP to hold (in particular Theorem 1), the following two conditions must be satisfied (see Lemma 2 and Lemma 3 in Appendix A):

1. The steady state $s_*$ (at the end of the first phase and at the beginning of the second phase) must satisfy the condition

$$s_* = F(x, s_*, \theta),$$
(52)

2. the Jacobian of the transition function $F$ must be symmetric, i.e.

$$\frac{\partial F}{\partial s}(x, s, \theta)^\top = \frac{\partial F}{\partial s}(x, s, \theta).$$
(53)

The condition of Eq. (53) is equivalent to the existence of a scalar function $\Phi(x, s, \theta)$ such that Eq. (51) holds. Going from Eq. (51) to Eq. (53) is straightforward: in this case the Jacobian of $F$ is the Hessian of $\Phi$, which is symmetric. Indeed $\frac{\partial F}{\partial s}(x, s, \theta) = \frac{\partial^2 \Phi}{\partial s^2}(x, s, \theta) = \frac{\partial F}{\partial s}(x, s, \theta)^\top$. Going from Eq. (53) to Eq. (51) is also true – though less obvious – and is a consequence of Green's theorem. [10] We say that $F$ derives from the scalar function $\Phi$, or that $\Phi$ is a primitive of $F$. Hence the name of 'primitive function' for $\Phi$.

**Assumption of Convergence in the Discrete-Time Setting.** In the real-time setting the gradient dynamics of Eq. 46 guarantees convergence to a (local) minimum of $E$. In contrast, in the discrete-time setting, no intrinsic property of $F$ or $\Phi$ a priori guarantees that the dynamics of Eq 47 settles to steady state. This discussion is out of the scope of this work and we refer to Scarselli et al. [2009] where sufficient (but not necessary) conditions are discussed to ensure convergence based on the contraction map theorem.

## B.3  Index Shift in the Definition of $\nabla_\theta^{\mathrm{BPTT}}(t)$ and $\Delta_\theta^{\mathrm{EP}}(t)$

The convention that we have chosen in Appendix A to define $\nabla_\theta^{\mathrm{BPTT}}(t)$ and $\Delta_\theta^{\mathrm{EP}}(t)$ could seem strange at first glance for two reasons:

- the state update $\Delta_s^{\mathrm{EP}}(t)$ is defined in terms of $s_t^\beta$ and $s_{t+1}^\beta$, whereas the weight update $\Delta_\theta^{\mathrm{EP}}(t)$ is defined in terms of $s_{t-1}^\beta$ and $s_t^\beta$,

- at time $t = 0$, the state gradient $\nabla_s^{\mathrm{BPTT}}(0)$ and the state update $\Delta_s^{\mathrm{EP}}(0)$ are defined, but the weight gradient $\nabla_\theta^{\mathrm{BPTT}}(0)$ and the weight update $\Delta_\theta^{\mathrm{EP}}(0)$ are not defined.

Here we explain why – when we dive deeper in the technical details – the convention adopted in Appendix A is in fact more natural than the one adopted in Sections 2.2 and 3.

First, recall from Appendix A.2 that we have defined the gradients of BPTT as

$$\forall t = 0, 1, \ldots, T, \qquad \nabla_s^{\mathrm{BPTT}}(t) = \frac{\partial \mathcal{L}}{\partial s_{T-t}}, \tag{54}$$

$$\forall t = 1, 2, \ldots, T, \qquad \nabla_\theta^{\mathrm{BPTT}}(t) = \frac{\partial \mathcal{L}}{\partial \theta_{T-t}}, \tag{55}$$

where

$$\forall t = 0, 1, \ldots T - 1, \qquad s_{t+1} = F(x, s_t, \theta_t = \theta), \qquad \mathcal{L} = \ell(s_T, y), \tag{56}$$

and from Appendix A.3 that we have defined the neural and weight updates of EP as

$$\forall t \geq 0, \qquad \Delta_s^{\mathrm{EP}}(t) = \lim_{\beta \to 0} \frac{1}{\beta}\left(s_{t+1}^\beta - s_t^\beta\right), \tag{57}$$

$$\forall t \geq 1, \qquad \Delta_\theta^{\mathrm{EP}}(t) = \lim_{\beta \to 0} \frac{1}{\beta}\left(\frac{\partial \Phi}{\partial \theta}\left(x, s_t^\beta, \theta\right) - \frac{\partial \Phi}{\partial \theta}\left(x, s_{t-1}^\beta, \theta\right)\right), \tag{58}$$

where

$$s_0^\beta = s_*, \qquad \forall t \geq 0, \quad s_{t+1}^\beta = F\left(x, s_t^\beta, \theta\right) - \beta\,\frac{\partial \ell}{\partial s}\left(s_t^\beta, y\right). \tag{59}$$

### B.3.1  Index Shift

Let us introduce

$$\Phi^\beta(x, s, y, \theta) = \Phi(x, s, \theta) - \beta\,\ell(s, y), \tag{60}$$

so that the dynamics in the second phase rewrites

$$s_{t+1}^\beta = \frac{\partial \Phi^\beta}{\partial s}\left(x, s_t^\beta, y, \theta\right). \tag{61}$$

It is then readily seen that the neural updates $\Delta_s^{\mathrm{EP}}$ and the weight updates $\Delta_\theta^{\mathrm{EP}}$ both rewrite in the form

$$\Delta_s^{\mathrm{EP}}(0) = \lim_{\beta \to 0} \frac{1}{\beta} \left( \frac{\partial \Phi^\beta}{\partial s} \left( x, s_0^\beta, y, \theta \right) - \frac{\partial \Phi}{\partial s} \left( x, s_0^\beta, \theta \right) \right), \tag{62}$$

$$\forall t \geq 1, \qquad \Delta_s^{\mathrm{EP}}(t) = \lim_{\beta \to 0} \frac{1}{\beta} \left( \frac{\partial \Phi^\beta}{\partial s} \left( x, s_t^\beta, y, \theta \right) - \frac{\partial \Phi^\beta}{\partial s} \left( x, s_{t-1}^\beta, y, \theta \right) \right), \tag{63}$$

$$\forall t \geq 1, \qquad \Delta_\theta^{\mathrm{EP}}(t) = \lim_{\beta \to 0} \frac{1}{\beta} \left( \frac{\partial \Phi^\beta}{\partial \theta} \left( x, s_t^\beta, y, \theta \right) - \frac{\partial \Phi^\beta}{\partial \theta} \left( x, s_{t-1}^\beta, y, \theta \right) \right). \tag{64}$$

Written in this form, we see a symmetry between $\Delta_s^{\mathrm{EP}}(t)$ and $\Delta_\theta^{\mathrm{EP}}(t)$ and there is no more index shift.

### B.3.2 Missing Weight Gradient $\nabla_\theta^{\mathrm{BPTT}}(0)$ and Weight Update $\Delta_\theta^{\mathrm{EP}}(0)$

We can naturally extend the definition of $\nabla_\theta^{\mathrm{BPTT}}(0)$ and $\Delta_\theta^{\mathrm{EP}}(0)$ following Eq. 55. In the setting studied in this paper, they both take the value 0 because the cost function $\ell(s, y)$ does not depend on the parameter $\theta$. But suppose now that $\ell$ depends on $\theta$, i.e. that $\ell$ is of the form $\ell(s, y, \theta)$. Then the loss of Eq. 56 takes the form $\mathcal{L} = \ell(s_T, y, \theta_T = \theta)$, so that:

$$\nabla_\theta^{\mathrm{BPTT}}(0) = \frac{\partial \mathcal{L}}{\partial \theta_T} = \frac{\partial \ell}{\partial \theta} (s_T, y, \theta). \tag{65}$$

As for the missing weight update $\Delta_\theta^{\mathrm{EP}}(0)$, we follow the definition of Eq. 62 and define:

$$\Delta_\theta^{\mathrm{EP}}(0) = \lim_{\beta \to 0} \frac{1}{\beta} \left( \frac{\partial \Phi^\beta}{\partial \theta} \left( x, s_0^\beta, y, \theta \right) - \frac{\partial \Phi}{\partial \theta} \left( x, s_0^\beta, \theta \right) \right) = -\frac{\partial \ell}{\partial \theta} (s_*, y, \theta). \tag{66}$$

Since $s_T = s_*$ (the state at the end of the first phase is the state at the beginning of the second phase, and it is the steady state), we have $\Delta_\theta^{\mathrm{EP}}(0) = -\nabla_\theta^{\mathrm{BPTT}}(0)$.

## C  Experiments: Demonstrating the GDU Property

### C.1  Definition of the Relative Mean Square Error (RelMSE)

We introduce a relative mean squared error (RelMSE) [11] between two continuous functions $f$ and $g$ in a given layer L as:

$$\text{RelMSE}(f, g) = \left\langle \frac{\|f - g\|_{2,K}}{\max(\|f\|_{2,K}, \|g\|_{2,K})} \right\rangle_L, \tag{67}$$

where $\|f\|_{2,K} = \sqrt{\frac{1}{K} \int_0^K f^2(t)dt}$ and $\langle \cdot \rangle_L$ denotes an average over all the elements of layer L. For example, $\text{RelMSE}(\Delta_{W_{01}}^{\text{EP}}, -\nabla_{W_{01}}^{\text{BPTT}})$ averages the squared distance between $\Delta_{W_{01}}^{\text{EP}}$ and $-\nabla_{W_{01}}^{\text{BPTT}}$ averaged over all the elements of $W_{01}$. Also, instead of computing $\Delta^{\text{EP}}$ and $\nabla^{\text{BPTT}}$ processes on a single sample presentation and bias the RelMSE by the choice of this sample, $\Delta^{\text{EP}}$ and $\nabla^{\text{BPTT}}$ processes have been averaged over a mini-batch of 20 samples before their distance in terms of RelMSE was measured.

### C.2  Details on section 4.2

Figure 6: Toy model architecture

**Equations.**  The toy model is an architecture where input, hidden and output neurons are connected altogether, without lateral connections. Denoting input neurons as $x$, hidden neurons as $s_1$ and output neurons as $s_0$, the primitive function for this model reads:

$$\Phi\left(x, s^0, s^1\right) = (1 - \epsilon)\frac{1}{2}\left(\|s^0\|^2 + \|s^1\|^2\right)$$
$$+ \epsilon\left(\sigma(s^0) \cdot W_{01} \cdot \sigma(s^1) + \sigma(s^0) \cdot W_{0x} \cdot \sigma(x) + \sigma(s^1) \cdot W_{1x} \cdot \sigma(x)\right),$$

where $\epsilon$ is a discretization parameter. Furthermore the cost function $\ell$ is

$$\ell\left(s^0, y\right) = \frac{1}{2}\|s^0 - y\|^2. \tag{68}$$

As a reminder, we define the following convention for the dynamics of the second phase: $\forall t \in [0, K]$ : $s_t^{n,\beta} = s_{t+T}^n$ where $T$ is the length of the first phase. The equations of motion read in the first phase read

$$\forall t \in [0, T] : \begin{cases} s_{t+1}^0 &= (1 - \epsilon)s_t^0 + \epsilon\sigma'(s_t^0)) \odot (W_{01} \cdot \sigma(s_t^1) + W_{0x} \cdot \sigma(x)) \\ s_{t+1}^1 &= (1 - \epsilon)s_t^1 + \epsilon\sigma'(s_t^1) \odot (W_{01}^\top \cdot \sigma(s_t^0) + W_{1x} \cdot \sigma(x)), \end{cases}$$

Table 3: Table of hyperparameters used to demonstrate Theorem 1. "EB" and "P" respectively denote "energy-based" and "prototypical", "-#h" stands for the number of hidden layers.

|  | Activation | T | K | $\beta$ | $\epsilon$ |
|---|---|---|---|---|---|
| Toy model | tanh | 5000 | 80 | 0.01 | 0.08 |
| EB-1h | tanh | 800 | 80 | 0.001 | 0.08 |
| EB-2h | tanh | 5000 | 150 | 0.01 | 0.08 |
| EB-3h | tanh | 30000 | 200 | 0.02 | 0.08 |
| P-1h | tanh | 150 | 10 | 0.01 | - |
| P-2h | tanh | 1500 | 40 | 0.01 | - |
| P-3h | tanh | 5000 | 40 | 0.015 | - |
| P-conv | hard sigmoid | 5000 | 10 | 0.02 | - |

In the second phase

$$\forall t \in [0, K] : \begin{cases} s_{t+1}^{0,\beta} &= (1 - \epsilon)s_t^{0,\beta} + \epsilon\sigma'(s_t^{0,\beta}) \odot (W_{01} \cdot \sigma(s_t^{1,\beta}) + W_{0x} \cdot \sigma(x)) \\ &\quad + \epsilon\beta(y - s_t^{0,\beta}) \\ s_{t+1}^{1,\beta} &= (1 - \epsilon)s_t^{1,\beta} + \epsilon\sigma'(s_t^{1,\beta}) \odot (W_{01}^\top \cdot \sigma(s_t^{0,\beta}) + W_{1x} \cdot \sigma(x)), \end{cases} \tag{69}$$

where $y$ denotes the target. In this case and according to the definition Eq. (10), the EP error processes for the parameters $\theta = \{W_{01}, W_{0x}, W_{1x}\}$ read:

$$\forall t \in [0, K] : \begin{cases} \Delta_{W_{01}}^{\mathrm{EP}}(t) &= \frac{1}{\beta}\left(\sigma(s_{t+1}^{0,\beta}) \cdot \sigma(s_{t+1}^{1,\beta})^\top - \sigma(s_t^{0,\beta}) \cdot \sigma(s_t^{1,\beta})^\top\right) \\ \Delta_{W_{0x}}^{\mathrm{EP}}(t) &= \frac{1}{\beta}\left(\sigma(s_{t+1}^{0,\beta}) \cdot \sigma(x)^\top - \sigma(s_t^{0,\beta}) \cdot \sigma(x)^\top\right) \\ \Delta_{W_{1x}}^{\mathrm{EP}}(t) &= \frac{1}{\beta}\left(\sigma(s_{t+1}^{1,\beta}) \cdot \sigma(x)^\top - \sigma(s_t^{1,\beta}) \cdot \sigma(x)^\top\right), \end{cases}$$

**Experiment: theorem demonstration on dummy data.** We took 5 output neurons, 50 hidden neurons and 10 visible neurons, using $\sigma(x) = \tanh(x)$. The experiment consists of the following: we define a dummy uniformly distributed random input $x \sim U[0, 1]$ (of size $1 \times 10$) and a dummy random one-hot encoded target (of size $1 \times 5$). We take $\epsilon = 0.08$ and perform the first phase for $T = 5000$ steps. Then, we perform on the one hand BPTT over $K = 80$ steps (to compute the gradients $\nabla^{\mathrm{BPTT}}$), on the other hand EP over $K = 80$ steps with $\beta = 0.01$ (to compute the neural updates $\Delta^{\mathrm{EP}}$) and compare the gradients and neural updates provided by the two algorithms. The resulting curves can be found in the main text (Fig. 2).

**C.3  Details on subsection 4.3: Definition of the fully connected layered model in the energy-based setting**

Figure 7: Layered architecture

**Equations.** The fully connected layered model is an architecture where the neurons are only connected between two consecutive layers. We denote neurons of the n-th layer as $s^n$ with $n \in [0, N-1]$. Layers are labelled in a backward fashion: $n = 0$ labels the output layer, $n = 1$ the first hidden starting from the output layer, and $n = N - 1$ the visible layer so that there are $N - 2$ hidden layers. As a reminder, we define the following convention for the dynamics of the second phase: $\forall t \in [0, K] : s_t^{n,\beta} = s_{t+T}^n$ where $T$ is the length of the first phase. The primitive function of this model is defined as:

$$\Phi\left(x, s^0, s^1, \ldots, s^N\right) = \frac{1}{2}(1 - \epsilon)\left(\sum_{n=1}^{N} ||s^n||^2\right) + \epsilon \sum_{n=0}^{N-1} \sigma(s^n) \cdot W_{nn+1} \cdot \sigma(s^{n+1}) \tag{70}$$

so that the equations of motion read:

$$\forall t \in [0, T] : \begin{cases} s_{t+1}^0 &= (1 - \epsilon)s_t^0 + \epsilon\sigma'(s_t^0)) \odot W_{01} \cdot \sigma(s_t^1) \\ s_{t+1}^n &= (1 - \epsilon)s_t^n + \epsilon\sigma'(s_t^n)) \odot (W_{nn+1} \cdot \sigma(s_t^{n+1}) + W_{n-1n}^\top \cdot \sigma(s_t^{n-1})) \quad \forall n \in [1, N-2] \end{cases}$$

$$\forall t \in [0, K] : \begin{cases} s_{t+1}^{0,\beta} &= (1 - \epsilon)s_t^{0,\beta} + \epsilon\sigma'(s_t^{0,\beta})) \odot W_{01} \cdot \sigma(s_t^{1,\beta}) + \beta\epsilon(y - s^{0,\beta}(t)) \\ s_{t+1}^{n,\beta} &= (1 - \epsilon)s_t^{n,\beta} + \epsilon\sigma'(s_t^{n,\beta})) \odot (W_{nn+1} \cdot \sigma(s_t^{n+1,\beta}) + W_{n-1n}^\top \cdot \sigma(s_t^{n-1,\beta})) \\ & \forall n \in [1, N-2] \end{cases} \tag{71}$$

In this case and according to the definition Eq. 10, the EP error processes for the parameters $\theta = \{W_{nn+1}\}$ read:

$$\forall t \in [0, K], \qquad \forall n \in [0, N-2] : \Delta_{W_{nn+1}}^{\mathrm{EP}}(t) = \frac{1}{\beta}\left(\sigma(s_{t+1}^{n,\beta}) \cdot \sigma(s_{t+1}^{n+1,\beta})^\top - \sigma(s_t^{n,\beta}) \cdot \sigma(s_t^{n+1,\beta})^\top\right)$$

**Experiment: theorem demonstration on MNIST.** For this experiment, we consider architectures of the kind 784-512-...-512-10 where we have 784 input neurons, 10 ouput neurons, and each hidden layer has 512 neurons, using $\sigma(x) = \tanh(x)$. The experiment consists of the following: we take a random MNIST sample (of size $1 \times 784$) and the associated target (of size $1 \times 10$). For a given $\epsilon$, we perform the first phase for $T = 2000$ steps. Then, we perform on the one hand BPTT over $K = $ steps (to compute the gradients $\nabla^{\mathrm{BPTT}}$), on the other hand EP over $K$ steps with a given $\beta$ (to compute the neural updates $\Delta^{\mathrm{EP}}$) and compare the gradients and neural updates provided by the two algorithms. Precise values of the hyperparameters $\epsilon$, T, K, beta are given in Tab. 3.

### C.4 Details on subsection 4.3: Fully connected layered architecture in the prototypical setting

**Equations.** The dynamics of the fully connected layered model are defined by the following set of equations:

$$\forall t \in [0, T] : \begin{cases} s_{t+1}^0 &= \sigma(W_{01} \cdot s_t^1) \\ s_{t+1}^n &= \sigma(W_{nn+1} \cdot s_t^{n+1} + W_{n-1n}^\top \cdot s_t^{n-1}) \quad \forall n \in [1, N-2] \end{cases}$$

$$\forall t \in [0, K] : \begin{cases} s_{t+1}^{0,\beta} &= \sigma(W_{01} \cdot s_t^{1,\beta}) + \beta(y - s^{0,\beta}(t)) \\ s_{t+1}^{n,\beta} &= \sigma(W_{nn+1} \cdot s_t^{n+1,\beta} + W_{n-1n}^\top \cdot s_t^{n-1,\beta}) \quad \forall n \in [1, N-2], \end{cases}$$

where y denotes the target. Considering the function:

$$\Phi\left(x, s^0, s^1, \ldots, s^N\right) = \sum_{n=0}^{N-1} s^n \cdot W_{nn+1} \cdot s^{n+1}, \tag{72}$$

and ignoring the activation function, we have:

$$s_t^n \approx \frac{\partial \Phi}{\partial s^n}(x, s_{t-1}^0, \cdots, s_{t-1}^{N-1}) \tag{73}$$

so that in this case, we define the EP error processes for the parameters $\theta = \{W_{nn+1}\}$ as:

$$\forall t \in [0, K], \qquad \forall n \in [0, N-2] : \Delta_{W_{nn+1}}^{EP}(t) = \frac{1}{\beta}\left(s_{t+1}^{n,\beta} \cdot s_{t+1}^{n+1,\beta^\top} - s_t^{n,\beta} \cdot s_{T+t}^{n+1,\beta^\top}\right)$$

**Experiment: theorem demonstration on MNIST.** The experimental protocol is the exact same as the one used on the fully connected layered architecture in the energy-based setting, using the same activation function $\sigma(x) = \tanh(x)$. Precise values of the hyperparameters $\epsilon$, $T$, $K$, beta are given in Tab. 3.

## C.5 Figures Demonstrating the GDU Property

Figure 8: Demonstrating the GDU property in the energy-based setting (as predicted by Theorem 1) with the fully connected layered architecture with one hidden layer on MNIST.

Figure 9: Demonstrating the GDU property in the energy-based setting (as predicted by Theorem 1) with the fully connected layered architecture with two hidden layers on MNIST.

Figure 10: Demonstrating the GDU property in the energy-based setting (as predicted by Theorem 1) with the fully connected layered architecture with three hidden layers on MNIST.

Figure 11: Demonstrating the GDU property in the prototypical setting (as predicted by Theorem 1) with the fully connected layered architecture with one hidden layer on MNIST.

Figure 12: Demonstrating the GDU property in the prototypical setting (as predicted by Theorem 1) with the fully connected layered architecture with two hidden layers on MNIST.

Figure 13: Demonstrating the GDU property in the prototypical setting (as predicted by Theorem 1) with the fully connected layered architecture with three hidden layers on MNIST.

## C.6 Why are the $\nabla_s^{\mathrm{BPTT}}$ and $\Delta_s^{\mathrm{EP}}$ processes saw teeth shaped in the prototypical setting ?

In the prototypical setting, in the case of a layered architecture (without lateral and skip-layer connections), the $\nabla^{\mathrm{BPTT}}$ and $\Delta^{\mathrm{EP}}$ processes are saw teeth shaped, i.e. they take the value zero every other time step (as seen per Fig. 4, Fig. 11, Fig. 12 and Fig. 13). We provide an explanation for this phenomenon both from the point of view of BPTT and from the point of view of EP. Fig. 14 illustrates this phenomenon in the case of a network with two layers: one output layer $s^0$ and one hidden layer $s^1$.

- **Point of view of BPTT.** In the forward-time pass (first phase), $s_{t+1}^0$ is determined by $s_t^1$, while $s_{t+1}^1$ is determined by $s_t^0$. This gives rise to a zig-zag shaped connectivity pattern in the computational graph of the the network unrolled in time (Fig. 14). In particular, the gray nodes of Fig. 14 are not involved in the computation of the loss $\mathcal{L}$, i.e. their gradients are equal to zero. In other words $\nabla_{s^1}^{\mathrm{BPTT}}(0) = 0$, $\nabla_{s^0}^{\mathrm{BPTT}}(1) = 0$, $\nabla_{s^1}^{\mathrm{BPTT}}(2) = 0$, etc.

- **Point of view of EP.** At the beginning of the second phase (at time step $t = 0$), the network is at the steady state ; in particular $s_0^{1,\beta} = s_*^1$. At time step $t = 1$, only the output layer $s^0$ is influenced by $y$ ; the hidden layer $s^1$ is still at the steady state, i.e. $s_1^{1,\beta} = s_*^1$. From $s_0^{1,\beta} = s_1^{1,\beta}$, it follows that $s_1^{0,\beta} = s_2^{0,\beta}$. In turn, from $s_1^{0,\beta} = s_2^{0,\beta}$ it follows that $s_2^{1,\beta} = s_3^{1,\beta}$. Etc. In other words $\Delta_{s^1}^{\mathrm{EP}}(0) = 0$, $\Delta_{s^0}^{\mathrm{EP}}(1) = 0$, $\Delta_{s^1}^{\mathrm{EP}}(2) = 0$, etc.

The above argument can be generalized to an arbitrary number of layers. In this case we group the layers of even index (resp. odd index) together. We call $e_t = \left(s_t^0, s_t^2, s_t^4, \ldots\right)$ and $o_t = \left(s_t^1, s^3, t, s_t^5, \ldots\right)$. The crucial property is that $o_{t+1}$ (resp. $e_{t+1}$) is determined by $e_t$ (resp. $o_t$).

One consequence of this analysis is that, in the prototypical setting of EP, half of the computations are redundant and could be avoided. Avoiding such redundant computations would lead to an implementation where the layers of even indices and the layers of odd indices are updated alternatively, similar to the one proposed in section 4.3 of Scellier and Bengio [2016].

Figure 14: Explanation of the saw teeth shape of the $\nabla_s^{\text{BPTT}}$ and $\Delta_s^{\text{EP}}$ processes in the prototypical setting (layered architecture without lateral or skip-layer connections). **Forward-time pass (top left)**: gray nodes in the computational graph indicate nodes that are not involved in the computation of the loss $\mathcal{L}$. **BPTT (bottom left)**: red arrows indicate the differentiation path through the output units $s^0$. The gradients in the gray nodes are equal to 0. **EP (bottom right)**: nodes of the same color have the same value.

In contrast, the saw teeth shaped curves are not observed in the energy based setting. This is due to the different topology of the computational graph in this setting. In the energy-based setting, the assumptions under which we have shown the saw teeth shape are not satisfied since neurons are subject to leakage, e.g. $s_{t+1}^1$ depends not just on $s_t^0$ but also on $s_t^1$. Therefore the reasoning developed above no longer holds.

### C.7 Demonstrating Theorem 1 on MNIST with the convolutional model.

The convolutional model is defined in Appendix D.

We have implemented an architecture with 2 convolution-pooling layers and 1 fully connected layer. The first and second convolution layers are made up of $5 \times 5$ kernels with 32 and 64 feature maps respectively. Convolutions are performed without padding and with stride 1. Pooling is performed with $2 \times 2$ filters and with stride 2.

The experimental protocol is the exact same as the one used on the fully connected layered architecture. The only difference is the activation function that we have used here is $\sigma(x) = \max(\min(x, 1), 0)$

which we shall refer to here for convenience as 'hard sigmoid function'. Precise values of the hyperparameters $\epsilon$, T, K, beta are given in Tab. 3.

We show on Fig. 4 that $\Delta^{\mathrm{EP}}$ and $-\nabla^{\mathrm{BPTT}}$ processes qualitatively very well coincide when presenting one MNIST sample to the network. Looking more carefully, we note that some $\Delta_s^{\mathrm{EP}}$ processes collapse to zero. This signals the presence of neurons which saturate to their maximal or minimal values, as an effect of the non linearity used. Consequently, as these neurons cannot move, they cannot carry the error signals. We hypothesize that this accounts for the discrepancy in the results obtained with EP on the convolutional architecture with respect to BPTT.

Figure 15: RelMSE analysis in the convolutional architecture.

# D Convolutional model (subsection 4.4)

Figure 16: Convolutional architecture. Summary of the operations, notations and conventions adopted in this section.

**Definition of the operations.** In this section, we define the following operations:

- the *convolution* of a filter W of size F with $C_{\text{out}}$ output channels and $C_{\text{in}}$ input channels by a vector X as:

$$(W * X)_{c_{\text{out}},i,j} := \sum_{c_{\text{int}}=1}^{C_{\text{in}}} \sum_{r,s=1}^{F} W_{c_{\text{out}},c_{\text{in}},r,s} X_{c_{\text{in}},i+r-1,j+s-1,}, \tag{74}$$

- the associated *transpose convolution* is defined as the convolution of kernel $\tilde{W}$ (also called "flipped kernel"):

$$\tilde{W}_{c_{in},c_{out},r,s} = W_{c_{out},c_{in},F-r+1,F-s+1}, \tag{75}$$

with an input padded with $\tilde{P} = F - 1 - P$ where P denotes the padding applied in the forward convolution: in this way transpose convolution recovers the original input size before convolution. Whenever $\tilde{W}$ is applied on a vector, we shall implicitly assume this padding.

- We define the *general dot product* between two vectors $X^1$ and $X^2$ as:

$$X^1 \bullet X^2 = \sum_{c_{\text{in}}=1}^{C_{\text{in}}} \sum_{i,j=1}^{d} X^1_{c_{\text{in}},i,j} X^2_{c_{\text{in}},i,j}. \tag{76}$$

- We define the *pooling* operation with filter size F and stride F as:

$$\mathcal{P}(X;F)_{c,i,j} = \max_{r,s \in [0,F-1]} \left\{ X_{c,F(i-1)+1+r,F(j-1)+1+s} \right\}. \tag{77}$$

We also introduce the relative indices within a pooling zone for which the maximum is reached as:

$$\text{ind}(X;F)_{c,i,j} = \underset{r,s \in [0,F-1]}{\arg\max} \left\{ X_{c,F(i-1)+1+r,F(j-1)+1+s} \right\} = (r^*(X,i), s^*(X,j)). \tag{78}$$

- We define the *inverse pooling* operation as:

$$\mathcal{P}^{-1}(Y, \text{ind}(X))_{c,p,q} = \begin{cases} Y_{c,\lceil p/F \rceil,\lceil q/F \rceil} \text{if} & p = F(\lceil p/F \rceil - 1) + 1 + r^*(X, \lceil p/F \rceil), \\ & q = F(\lceil q/F \rceil - 1) + 1 + s^*(X, \lceil q/F \rceil) \\ 0 & \text{otherwise} \end{cases} \tag{79}$$

In layman terms, the inverse pooling operation applied to a vector $Y$ given the indices of another vector $X$ up-samples $Y$ to a vector of the same size of $X$ with the elements of Y located at the maximal elements of $X$ within each pooling zone, and zero elsewhere.

Note that Eq. (79) can be written more conveniently as:

$$\mathcal{P}^{-1}(Y, \mathrm{ind}(X))_{c,p,q} = \sum_{i,j} Y_{c,i,j} \cdot \delta_{p,F(i-1)+1+r^*(X,i)} \cdot \delta_{q,F(j-1)+1+s^*(X,j)}. \tag{80}$$

- The *flattening* operation which maps a vector X into its flattened shape, i.e. $\mathcal{F} : C^{\mathrm{in}} \times D \times D \to 1 \times C^{\mathrm{in}} D^2$. We denote its inverse operation, i.e. the *inverse flattening operation* as $\mathcal{F}^{-1}$.

**Equations.** The model is a layered architecture composed of a fully connected part and a convolutional part. We therefore distinguish between the flat layers (i.e. those of the fully connected part) and the convolutional layers (i.e. those of the convolutional part). We denote $N_{\mathrm{fc}}$ and $N_{\mathrm{conv}}$ the number of flat layers and of convolutional layers respectively.

As previously, layers are labelled in a backward fashion: $s^0$ labels the output layer, $s^1$ the first hidden starting from the output layer (i.e. the first flat layer), and $s^{N_{\mathrm{fc}}-1}$ the last flat layer. Fully connected layers are bi-dimensional[12], i.e. $s_{i,j}$ where i and j label one pixel.

The layer $h^0$ denotes the first convolutional layer that is being flattened before being fed to the classifier part. From there on, $h^1$ denotes the second convolutional layer, $h^{N_{\mathrm{conv}}-1}$ the last convolutional layer and $h^{N_{\mathrm{conv}}}$ labels the visible layer. Convolutional layers are three-dimensional [13], i.e. $s_{c,i,j}$ where c labels a channel, i and j label one pixel of this channel.

A convolutional layer $h_n$ is deduced from an *upstream* convolutional layer $h_{n-1}$ by the composition of a convolution and a pooling operation, which we shall respectively denote by $*$ and $\mathcal{P}$. Conversely, a convolutional layer $h_n$ is deduced from a *downstream* convolutional layer $h_{n+1}$ by the composition of a unpooling operation and of a transpose convolution. We note $W^{\mathrm{fc}}$ and $W^{\mathrm{conv}}$ the fully connected weights and the convolutional filters respectively, so that $W^{\mathrm{fc}}$ is a two-order tensor and $W^{\mathrm{conv}}$ is a four order tensor, i.e. $W^{\mathrm{conv}}_{c_{out},c_{in},i,j}$ is the element $(i,j)$ of the feature map connecting the input channel $c_{in}$ to the output channel $c_{out}$. We denote the filter size by F. We keep the same notation $x$ for the input data.

With this set of notations, the equations in the fully connected layers read in the first phase:

$$\forall t \in [0,T] : \begin{cases} s_{t+1}^0 &= \sigma\left(W_{01}^{\mathrm{fc}} \cdot s_t^1\right) \quad \text{(output layer)} \\ s_{t+1}^n &= \sigma\left(W_{nn+1}^{\mathrm{fc}} \cdot s_t^{n+1} + W_{n-1n}^{\mathrm{fc}\top} \cdot s_t^{n-1}\right) \quad \forall n \in [1, N_{\mathrm{fc}}-2] \\ s_{t+1}^{N_{\mathrm{fc}}-1} &= \sigma\left(W_{N_{\mathrm{fc}}-1,N_{\mathrm{fc}}}^{\mathrm{fc}} \cdot \mathcal{F}(h_t^0) + W_{N_{\mathrm{fc}}-2,N_{\mathrm{fc}}-1}^{\mathrm{fc}\top} \cdot s_t^{N_{\mathrm{fc}}-2}\right) \quad \text{(last fully connected layer)} \end{cases},$$

and in the second phase:

$$\forall t \in [0,T] : \begin{cases} s_{t+1}^0 &= \sigma\left(W_{01}^{\mathrm{fc}} \cdot s_t^1\right) + \beta(y - s^0) \quad \text{(nudged output layer)} \\ s_{t+1}^n &= \sigma\left(W_{nn+1}^{\mathrm{fc}} \cdot s_t^{n+1} + W_{n-1n}^{\mathrm{fc}\top} \cdot s_t^{n-1}\right) \quad \forall n \in [1, N_{\mathrm{fc}}-2] \\ s_{t+1}^{N_{\mathrm{fc}}-1} &= \sigma\left(W_{N_{\mathrm{fc}}-1,N_{\mathrm{fc}}}^{\mathrm{fc}} \cdot \mathcal{F}(h_t^0) + W_{N_{\mathrm{fc}}-2,N_{\mathrm{fc}}-1}^{\mathrm{fc}\top} \cdot s_t^{N_{\mathrm{fc}}-2}\right) \quad \text{(last fully connected layer)} \end{cases},$$

where y denotes the target. Conversely, convolutional layers read the following set of equations at all time:

$$\forall t \in [0,T] : \begin{cases} h_{t+1}^0 &= \sigma\left(\mathcal{P}\left(W_{01}^{\mathrm{conv}} * h_t^1\right) + \mathcal{F}^{-1}\left(W_{N_{\mathrm{fc}}-1,N_{\mathrm{fc}}}^{\mathrm{fc}\top} \cdot s_t^{N_{\mathrm{fc}}-1}\right)\right) \quad \text{(first convolutional layer)} \\ h_{t+1}^n &= \sigma\left(\mathcal{P}\left(W_{n,n+1}^{\mathrm{conv}} * h_t^{n+1}\right) + \tilde{W}_{n-1,n}^{\mathrm{conv}} * \mathcal{P}^{-1}\left(h_t^{n-1}, \mathrm{ind}(W_{n-1,n}^{\mathrm{conv}} * h_{t-1}^n)\right)\right) \forall n \in [1, N_{\mathrm{conv}}-1] \end{cases},$$

where by convention $h^{N_{\mathrm{conv}}} = x$. From here on, we shall omit the second argument of inverse pooling $\mathcal{P}^{-1}$ - i.e. the locations of the maximal neuron values before applying pooling - to improve readability of the equations and proofs. Considering the function:

$$\Phi(x, s_0, \cdots, s_{N_{\text{fc}}-1}, h_0, \cdots, h_{N_{\text{fc}}-1}) = \sum_{n=0}^{N_{\text{fc}}-1} s^{n\top} \cdot W_{n,n+1}^{\text{fc}} \cdot s^{n+1} + s^{N_{\text{fc}}-1} \cdot W_{N_{\text{fc}}-1, N_{\text{fc}}}^{\text{fc}} \cdot \mathcal{F}(h_t^0)$$

$$+ \sum_{n=1}^{N_{\text{conv}}-1} h^n \bullet \mathcal{P}\left(W_{n,n+1}^{\text{conv}} * h^{n+1}\right),$$

and ignoring the activation function, we have:

$$\begin{cases} \forall n \in [0, N_{\text{fc}} - 1]: & s_t^n \approx \frac{\partial \Phi}{\partial s^n}(x, s_0, \cdots, s_{N_{\text{fc}}-1}, h_0, \cdots, h_{N_{\text{fc}}-1}) \\ \forall n \in [0, N_{\text{conv}} - 1]: & h_t^n \approx \frac{\partial \Phi}{\partial h^n}(x, s_0, \cdots, s_{N_{\text{fc}}-1}, h_0, \cdots, h_{N_{\text{fc}}-1}) \end{cases}, \quad (81)$$

so that in this case, we define the EP error processes for the parameters $\theta = \{W_{nn+1}^{\text{fc}}, W_{nn+1}^{\text{conv}}\}$ as:

$$\forall t \in [0, K], \forall n \in [0, N_{\text{fc}} - 2]: \quad \Delta_{W_{nn+1}^{\text{fc}}}^{\text{EP}}(t) = \frac{1}{\beta}\left(s_{T+t+1}^n \cdot s_{T+t+1}^{n+1\top} - s_{T+t}^n \cdot s_{T+t}^{n+1\top}\right)$$

$$\forall t \in [0, K]: \quad \Delta_{W_{N_{\text{fc}}-1, N_{\text{fc}}}^{\text{fc}}}^{\text{EP}}(t) = \frac{1}{\beta}\left(s_{T+t+1}^{N_{\text{fc}}-1} \cdot \mathcal{F}\left(h_{T+t+1}^0\right)^\top - s_{T+t}^{N_{\text{fc}}-1} \cdot \mathcal{F}\left(h_{T+t}^0\right)^\top\right)$$

$$\forall t \in [0, K], \forall n \in [0, N_{\text{conv}}-2]: \quad \Delta_{W_{nn+1}^{\text{conv}}}^{\text{EP}}(t) = \frac{1}{\beta}\left(\mathcal{P}^{-1}(h_{T+t+1}^n) * h_{T+t+1}^{n+1} - \mathcal{P}^{-1}(h_{T+t}^n) * h_{T+t}^{n+1}\right)$$

$$(82)$$

To further justify Eq. (81) and Eq. (82), we state and prove the following lemma.

**Lemma 5.** *Taking:*

$$\Phi = Y \bullet \mathcal{P}\left(W * X\right),$$

*and denoting $Z = W * X$, we have:*

$$\frac{\partial \Phi}{\partial Z} = \mathcal{P}^{-1}\left(Y\right) \tag{83}$$

$$\frac{\partial \Phi}{\partial X} = \tilde{W} * \mathcal{P}^{-1}\left(Y\right) \tag{84}$$

$$\frac{\partial \Phi}{\partial W} = \mathcal{P}^{-1}\left(Y\right) * X \tag{85}$$

$$\frac{\partial \Phi}{\partial Y} = \mathcal{P}\left(W * X\right) \tag{86}$$

*Proof of Lemma 5.* Let us prove Eq. (83). We have:

$$\frac{\partial \Phi}{\partial Z_{c,x,y}} = \sum_{c',i,j} Y_{c',i,j} \frac{\partial \mathcal{P}(Z)_{c',i,j}}{\partial Z_{c,x,y}}$$

$$= \sum_{c',i,j} Y_{c',i,j} \frac{\partial Z_{c', F(i-1)+1+r^*(i), F(j-1)+1+s^*(j)}}{\partial Z_{c,x,y}}$$

$$= \sum_{i,j} Y_{c,i,j} \delta_{x, F(i-1)+1+r^*(i)} \delta_{y, F(j-1)+1+s^*(j)}$$

$$= \mathcal{P}^{-1}(Y)_{c,x,y},$$

where we used Eq. (80) at the last step.

We can now proceed to proving Eq. (84). We have:

$$\frac{\partial \Phi}{\partial X_{c,p,q}} = \sum_{c',x,y} \frac{\partial \Phi}{\partial Z_{c',x,y}} \cdot \frac{\partial Z_{c',x,y}}{\partial X_{c,p,q}}$$

$$= \sum_{c',x,y} \mathcal{P}^{-1}(Y)_{c',x,y} \cdot \frac{\partial}{\partial X_{c,p,q}} \left( \sum_{c'',r,s} W_{c',c'',r,s} X_{c'',x+r-1,y+s-1} \right)$$

$$= \sum_{c',x,y} \sum_{r,s} \mathcal{P}^{-1}(Y)_{c',x,y} W_{c',c,r,s} \delta_{p,x+r-1} \delta_{q,y+s-1}$$

$$= \sum_{c',r,s} W_{c',c,r,s} \mathcal{P}^{-1}(Y)_{c',p-(r-1),q-(s-1)}.$$

Using the flipped kernel $\tilde{W}$ and performing the change of variable $r \leftarrow F - r + 1$ and $s \leftarrow F - s + 1$, we obtain:

$$\frac{\partial \Phi}{\partial X_{c,p,q}} = \sum_{c',r,s} \tilde{W}_{c,c',r,s} \cdot \mathcal{P}^{-1}(Y)_{c',p+r-F,q+s-F}. \tag{87}$$

Note in Eq. (87) that $\mathcal{P}^{-1}(Y)$ indices can exceed their boundaries. Also, as stated previously, $\mathcal{P}^{-1}(Y)$ should be padded with $\tilde{P} = F - 1 - P$ so that we recover the size of X after transpose convolution. Without loss of generality, we assume $P = 0$. We subsequently defined the padded input $\overline{\mathcal{P}^{-1}(Y)}$ as:

$$\overline{\mathcal{P}^{-1}(Y)}_{c,p,q} = \begin{cases} \mathcal{P}^{-1}(Y)_{c,p-F+1,q-F+1} \text{ if } p,q \in [F, N+F-1] \\ 0 \text{ if } p,q \in [1, F-1] \cup [N+F, N+2(F-1)] \end{cases}, \tag{88}$$

where N denotes the dimension of $\mathcal{P}^{-1}(Y)$. Finally Eq. (87) can conveniently be rewritten as:

$$\frac{\partial \Phi}{\partial X_{c,p,q}} = \left( \tilde{W} * \overline{\mathcal{P}^{-1}(Y)} \right)_{p,q}. \tag{89}$$

For the sake of readability, the padding is implicitly assumed whenever transpose convolution is performed so that we drop the bar notation.

We can now proceed to proving Eq. (85). We have:

$$\frac{\partial \Phi}{\partial W_{c',c,r,s}} = \sum_{c'',x,y} \frac{\partial \Phi}{\partial Z_{c'',x,y}} \cdot \frac{\partial Z_{c'',x,y}}{\partial W_{c',c,r,s}}$$

$$= \sum_{c'',x,y} \mathcal{P}^{-1}(Y)_{c'',x,y} \cdot \frac{\partial}{\partial W_{c',c,r,s}} \left( \sum_{k,r',s'} W_{c'',k,r',s'} X_{k,x+r'-1,y+s'-1} \right)$$

$$= \sum_{x,y} \mathcal{P}^{-1}(Y)_{c',x,y} \cdot X_{c,r+x-1,s+y-1}$$

$$= \left( \mathcal{P}^{-1}(Y) * X \right)_{c',c,r,s}$$

Finally, proving Eq. (86) is straightforward.

$\square$

# E   Training Experiments (Table 1)

**Simulation framework.**   Simulations have been carried out in Pytorch. The code has been attached to the supplementary materials upon submitting this work on the CMT interface. We have also attached a readme.txt with a specification of all dependencies, packages, descriptions of the python files as well as the commands to reproduce all the results presented in this paper.

**Data set.**   Training experiments were carried out on the MNIST data set. Training set and test set include 60000 and 10000 samples respectively.

**Optimization.**   Optimization was performed using stochastic gradient descent with mini-batches of size 20. For each simulation, weights were Glorot-initialized. No regularization technique was used and we did not use the persistent trick of caching and reusing converged states for each data sample between epochs as in [Scellier and Bengio, 2017].

**Hyperparameter search for EP.**   We distinguish between two kinds of hyperparameters: the recurrent hyperparameters - i.e. $T$, $K$ and $\beta$ - and the learning rates. A first guess of the recurrent hyperparameters $T$ and $\beta$ is found by plotting the $\Delta^{\mathrm{EP}}$ and $\nabla^{\mathrm{BPTT}}$ processes associated to synapses and neurons to see qualitatively whether the theorem is approximately satisfied, and by conjointly computing the proportions of synapses whose $\Delta_W^{\mathrm{EP}}$ processes have the same sign as its $\nabla_W^{\mathrm{BPTT}}$ processes. $K$ can also be found out of the plots as the number of steps which are required for the gradients to converge. Morever, plotting these processes reveal that gradients are vanishing when going away from the output layer, i.e. they lose up to $10^{-1}$ in magnitude when going from a layer to the previous (i.e. upstream) layer. We subsequently initialized the learning rates with increasing values going from the output layer to upstreams layers. The typical range of learning rates is $[10^{-3}, 10^{-1}]$, $[10, 1000]$ for T, $[2, 100]$ for K and $[0.01, 1]$ for $\beta$. Hyperparameters where adjusted until having a train error the closest to zero. Finally, in order to obtain minimal recurrent hyperparameters - i.e. smallest T and K possible, both in the energy-based and prototypical setting for a fair comparison - we progressively decreased T and K until the train error increases again.

**Activation functions, update clipping.**   For training, we used two kinds of activation functions:

- $\sigma(x) = \frac{1}{1+\exp(-4(x-1/2))}$. Although it is a shifted and rescaled sigmoid function, we shall refer to this activation function as 'sigmoid'.
- $\sigma(x) = \max(\min(x, 1), 0)$. It is the 'hard' version of the previous activation function so that we call it here for convenience 'hard sigmoid'.

The sigmoid function was used for all the training simulations except the convolutional architecture for which we used the hard sigmoid function - see Table 4. Also, similarly to [Scellier and Bengio, 2017], for the energy-based setting we clipped the neuron updates between 0 and 1 so that at each time step, when an update $\Delta s$ was prescribed, we have implemented: $s \leftarrow \max(\min(s + \Delta s, 1), 0)$.

**Benchmarking EP with respect to BPTT.**   In order to compare EP and BPTT directly, for each simulation trial we used the same weight initialization to train the network with EP on the one hand, and with BPTT on the other hand. We also used the same learning rates, and the same recurrent hyperparameters: we used the same $T$ for both algorithms, and we truncated BPTT to $K$ steps, as prescribed by the theory.

Table 4: Table of hyperparameters used for training. "EB" and "P" respectively denote "energy-based" and "prototypical", "-#h" stands for the number of hidden layers.

|  | Activation | T | K | $\beta$ | $\epsilon$ | Epochs | Learning rates |
|---|---|---|---|---|---|---|---|
| EB-1h | sigmoid | 100 | 12 | 0.5 | 0.2 | 30 | 0.1-0.05 |
| EB-2h | sigmoid | 500 | 40 | 0.8 | 0.2 | 50 | 0.4-0.1-0.01 |
| P-1h | sigmoid | 30 | 10 | 0.1 | - | 30 | 0.08-0.04 |
| P-2h | sigmoid | 100 | 20 | 0.5 | - | 50 | 0.2-0.05-0.005 |
| P-3h | sigmoid | 180 | 20 | 0.5 | - | 100 | 0.2-0.05-0.01-0.002 |
| P-conv | hard sigmoid | 200 | 10 | 0.4 | - | 40 | 0.15-0.035-0.015 |

---

**Algorithm 1** Discrete-time Equilibrium Propagation (EP)

---

*Input*: static input $x$, parameter $\theta$, learning rate $\alpha$.
*Output*: parameter $\theta$.

1: **while** $\theta$ not converged **do**
2:　　**for** each mini-batch x **do**
3:　　　　$\Delta\theta \leftarrow 0$
4:　　　　**for** $t \in [1, T]$ **do**
5:　　　　　　$s_{t+1} \leftarrow \frac{\partial \Phi}{\partial s}(x, s_t, \theta)$　　　　　　　　　$\triangleright$ 1$^{\text{st}}$ phase: common to EP and BPTT
6:　　　　**end for**
7:　　　　**for** $t \in [1, K]$ **do**
8:　　　　　　$s_{t+1}^{\beta} \leftarrow \frac{\partial \Phi^{\beta}}{\partial s}(x, s_t, \theta)$　　　　　　　$\triangleright$ 2$^{\text{nd}}$ phase: *forward-time* computation
9:　　　　　　$\Delta_{\theta}^{\text{EP}} \leftarrow \frac{1}{\beta}\left(\frac{\partial \Phi}{\partial \theta}(x, s_{t+1}^{\beta}, \theta) - \frac{\partial \Phi}{\partial \theta}(x, s_t^{\beta}, \theta)\right)$
10:　　　　　　$\Delta\theta \leftarrow \Delta\theta + \Delta_{\theta}^{\text{EP}}$
11:　　　　**end for**
12:　　　　$\theta \leftarrow \theta + \alpha\Delta\theta$
13:　　**end for**
14: **end while**

---

---

**Algorithm 2** Backpropagation Through Time (BPTT)

---

*Input*: static input $x$, parameter $\theta$, learning rate $\alpha$.
*Output*: parameter $\theta$.

1: **while** $\theta$ not converged **do**
2:　　**for** each mini-batch x **do**
3:　　　　$\Delta\theta \leftarrow 0$
4:　　　　**for** $t \in [1, T]$ **do**
5:　　　　　　$s_{t+1} \leftarrow \frac{\partial \Phi}{\partial s}(x, s_t, \theta)$　　　　　　　　　$\triangleright$ 1$^{\text{st}}$ phase: common to EP and BPTT
6:　　　　**end for**
7:　　　　**for** $t \in [1, K]$ **do**
8:　　　　　　$\nabla_{\theta}^{\text{BPTT}} \leftarrow \frac{\partial \mathcal{L}}{\partial \theta_{T-t}}$　　　　　　　　$\triangleright$ 2$^{\text{nd}}$ phase: *backward-time* computation
9:　　　　　　$\Delta\theta \leftarrow \Delta\theta + \nabla_{\theta}^{\text{BPTT}}$
10:　　　　**end for**
11:　　　　$\theta \leftarrow \theta - \alpha\Delta\theta$
12:　　**end for**
13: **end while**

---

Figure 17: Train and test error achieved on MNIST by the fully connected layered architecture with one hidden layer (784-512-10) in the energy-based setting throughout learning, over five trials. Plain lines indicate mean, shaded zones delimiting mean plus/minus standard deviation.

Figure 18: Train and test error achieved on MNIST by the fully connected layered architecture with two hidden layers (784-512-512-10) in the energy-based setting throughout learning, over five trials. Plain lines indicate mean, shaded zones delimiting mean plus/minus standard deviation.

Figure 19: Train and test error achieved on MNIST by the fully connected layered architecture with one hidden layer (784-512-10) in the prototypical setting throughout learning, over five trials. Plain lines indicate mean, shaded zones delimiting mean plus/minus standard deviation.

Figure 20: Train and test error achieved on MNIST by the fully connected layered architecture with two hidden layers (784-512-512-10) in the prototypical setting throughout learning, over five trials. Plain lines indicate mean, shaded zones delimiting mean plus/minus standard deviation.

Figure 21: Train and test error achieved on MNIST by the fully connected layered architecture with three hidden layers (784-512-512-512-10) in the prototypical setting throughout learning, over five trials. Plain lines indicate mean, shaded zones delimiting mean plus/minus standard deviation.

Figure 22: Train and test error achieved on MNIST by the convolutional architecture in the prototypical setting throughout learning, over five trials. Plain lines indicate mean, shaded zones delimiting mean plus/minus standard deviation.

## Footnotes

[7]Note that the stability of the steady state implies that the eigenvalues of the Jacobian $\frac{\partial F}{\partial s}(x, s_*, \theta)$ are smaller than 1 in magnitude. As a consequence of Lemma 2, the gradients $\nabla_\theta^{\mathrm{BPTT}}(t)$ decay (i.e. vanish) exponentially fast, which ensures that the full gradient $\sum_{t=0}^{K-1} \nabla_\theta^{\mathrm{BPTT}}(t)$ converges, even if $K \to \infty$. In the context of convergent RNNs with a static input, vanishing gradients of BPTT are consequently not a problem, as it is the case when learning from temporal data with RNNs.

[8]Note that we choose here a different convention for the definition of $\theta_t$ compared to the definition of Section 2.2. We motivate this index shift in Appendix B.3.

[9]Note the index shift in the definition of $\Delta_\theta^{\mathrm{EP}}(\beta, t)$ compared to the definition of Eq. 10. We motivate this index shift in Appendix B.3.

[10] Another equivalent formulation is that the curl of $F$ is null, i.e. $\overrightarrow{\mathrm{rot}}\,\vec{F} = \vec{0}$.

[11] We choose the RelMSE metric rather than a more conventional one such as the cos metric. Indeed, although the cos metric is also meaningful, it lacks an important property in our context: the cos between $f$ and $g$ is maximal if and only if $f$ and $g$ are proportional, whereas we aim at reaching equality (Theorem 1). In contrast, our RelMSE metric is such that $\text{RelMSE}(f, g) = 0 \Leftrightarrow f(t) = g(t)$.

[12]Three-dimensional in practice, considering the mini-batch dimension.

[13]Four-dimensional in pratice, considering the mini-batch dimension.