[Reviews · NeurIPS 2019]

Reviewer 1



The manuscript describes a discrete time reduction of equilibrium prop (EP) which enables the authors to compare the algorithms gradient estimates and performance directly to BPTT. Moreover, the associated reduction in compute cost also enables them to train the first CNN (that I know of) using EP. While EP approximates BP in feedforward networks, it uses neuronal activity of an RNN in equilibrium to propagate target information or error feedback to perform credit assignment. While this work may be less interesting for DL practitioners because it is still more costly than backprop (BP), it is one of the contenders for bio-plausible backprop which is discussed in the literature. In that regard the present work contributes to this discussion meaningfully. The paper is relatively well written, but still contains the occasional typo and generally the language could be improved. Overall, however, the messages are clear and well presented. I only have minor points which could be used to further improve the manuscript. Use of RMSE: I am wondering whether it would be better to use cos alpha between the EP and BP gradient to illustrate performance on all the plots. RMSE would be also susceptible to the length of the vector which can always be absorbed in the learning rate. Since here the authors seem to be mostly interested in the direction, something that agnostic to the length would seem more suitable. I found the formulation of the loss as a sum of losses a bit weird (ll.86). It would make more sense to me to put the temporal index on the loss function and not the parameters (since they are supposed to be same if I understood the authors correctly). Section 3.2: When comparing EP to BPTT, wouldn't it make more sense to compare to both BPTT & RTRL and find an umbrella term for both? RTRL will give the same gradient as BPTT, but somehow the manuscript makes the argument that EP is forward in time whereas BP isn't. But then RTRL is too ... so this is argument needs to be honed a bit. In practice, however, it is clear that gradients are computed using BPTT in time, I am only suggesting to amend the text here. Figure2 & Figure 4: I suggest changing the order of the plotted curves such that the dashed lines are somehow visible. That or play with transparency. Currently due to the overlap dashed is virtually invisible which is confusing. Finally, the reduction in computational cost is not shown which is one of the sales arguments for the discrete time version. However, it would be nice to add a small graph or table with some numbers in terms of wall clock time. l.41. "real-time leaky integrate neuronal dynamics" something is wrong with this sentence l.58. "similar performance than" -> to Update: Thanks for the clarifications on the RMSE and for adding the wall clock time. Finally, the updated figures, will further improve the MS. Thanks!

Reviewer 2



Even if the practical applicability of this learning algorithm on current hardware is limited, the theoretical approach and its derivation is certainly relevant to the NIPS community. The paper is well structured, the mathematical notation is well understandable and clear. Nevertheless I have some (minor) concerns. I miss a clear presentation of the restrictions on the transition function F and the role of the convergence “assumption” of the first phase. As far as I understood convergence of the first phase requires F’s such that F=d Phi /ds. The propotypical setting seams to state the same with other words, isn’t it. A clear relation between the fixed-point search and Energy-maximization might be obvious in the whole context of EP, but it is not clear enough from this paper. A discretized version of EP has to be compared to standard RNN approaches, hence also the relation to other non fixedpoint-converging, standard RNNs should be discussed. In particular a comparison with LSTM and a comment on the relation with the vanishing/exploding gradient problem and why this is not a problem in view of the fixedpoint search would be nice UPDATE: Thanks for the additional information in the rebuttal and the clarifications. Congrats to this excellent paper!

Reviewer 3



The authors provide a formulation of equilibrium propagation that is applicable to discrete time models that usually use backpropagation-through-time. The main theoretical result of the paper is that the updates from EP are equal on a step-by-step level with the updates of BPTT when the transition function of the RNN derives from the primitive function'' (which is similar to the energy function used in Scellier and Bengio 2019), and the RNN converges to a steady state over time. The authors further demonstrate this in practice, also for standard RNNs where this condition is not met. Originality: The results of this work seem to be original. Quality: The quality of the paper is high. The main theoretical results are explained in a clear and intuitive way, and the experiments are well-motivated. Clarity: The clarity of the paper is high. The setup, assumptions and theoretical results are clearly explained. The experiments are also well explained. The figures could benefit from having titles/legends or text in the caption clarifying each setup -- example in fig. 3, unless I read the main text, it's not clear what each column is. Another minor fix for clarity would be to explicitly specify what the weight updates are for each of the experiment, and when the weights are updated. The theory section could benefit from a brief verbal outline of the proof. Significance: The results of the paper are significant, since it elucidates how and when equilibrium prop algorithm would work for the discrete time case and for standard RNN models. The fact that the performance comes so close to BPTT on MNIST contributes to the significance. Some factors that could affect general applicability of this result is the use of symmetric weights, and the requirement of the network to converge quickly to a steady state. Further comments: - It would be interesting to see the behaviour of the algorithm when the weights are not symmetric. Is it expected to work? What would be the trajectory of the weights? - Quantitative data of how the performance degrades with more layers could help define the limits of the algorithm in terms of what the deepest networks this algorithm could train.

[Author Response · NeurIPS 2019]

We thank the reviewers for their time, thorough reading of our paper and extremely useful comments that we much appreciated reading. We address the points that were raised below.

**Reviewer 1**. We have now improved figure quality, playing with transparency of the plain lines so that the dashed lines are now visible. We also have computed the wall-clock time of each algorithm on MNIST training and added it as an extra column to Table 1 to highlight the benefits of our discrete-time setting (P) compared to the real-time setting (EB), where #-h stands for the number of hidden layers used: EB-1h: 1 hrs 33 mins, EB-2h: 16 hrs 4 mins, P-1h: 17 mins, P-2h: 1 hr 56 mins, P-3h: 8 hrs 27 mins, P-conv: 8 hrs 58 mins. To emphasize these results in the conclusion, we have amended, l. 205: "no degradation of accuracy is seen between using the prototypical (P) rather than the energy-based (EB) setting, although the prototypical setting requires three to five times less time steps in the first phase (T) and cuts the simulation time by a factor five to eight."

Our RMSE(f,g) metric, defined in Appendix C.2.3, is normalized by the max of $||f||$ and $||g||$ so that it is invariant upon rescaling functions f and g. Our RMSE is therefore not the usual Root Mean Squared Error, and stands for "Relative Mean Squared Error". We clarified this by changing our acronym to RelMSE. The suggested cos metric is also meaningful, but lacks an important property in our context: the cos between $f$ and $g$ is maximal if and only if $f$ and $g$ are proportional, whereas we aim at reaching equality (Theorem 1). In contrast, our RelMSE metric is such that $\text{RelMSE}(f, g) = 0 \Leftrightarrow f(t) = g(t)$.

We also bring a precision about the notation $\partial \mathcal{L}/\partial \theta_t$. We chose this notation (the index $t$ must be on $\theta$, not on $\mathcal{L}$) to highlight the difference with RNNs trained with sequential data. In this situation, the total loss reads $\mathcal{L} = \sum_t \mathcal{L}_t$ with the sum index carried by the losses, and so does the gradient of the total loss. However in our setting: $\mathcal{L}_t = 0$ for $t < T$, so that $\mathcal{L} = \mathcal{L}_T$. Then, the gradient $\partial \mathcal{L}_T / \partial \theta$ can itself be decomposed as $\partial \mathcal{L}_T / \partial \theta = \sum_{k=1}^T \mathcal{L}_T / \partial \theta_k$, with the sum index being carried by the parameters. We have added a clarification in our revised manuscript about this notation choice, in the paragraph preceding Eq.4.

Concerning RTRL, as suggested, we have amended the text, l. 124: "Other algorithms such as RTRL and UORO [Tallec and Ollivier, 2017] also compute the gradients by forward-time dynamics". Finally we have corrected, l. 41: "Originally, EP was introduced in the context of leaky-integrate neurons [Scellier and Bengio, 2017, 2019]. Computing their dynamics involves long simulation times, hence limiting EP training experiments to small neural networks".

**Reviewer 3**. To clarify the role of the conditions to be satisfied by the transition function F and as also suggested by Reviewer 4, we have now added the following verbal outline for Theorem 1, l. 118 just before the theorem statement: "The convergence requirement enables to derive the equations satisfied by the neural and weight updates (Lemma 4). Then, the existence of a primitive function ensures that these equations are equal to those satisfied by the gradients of BPTT (Corollary 3), with same initial conditions, yielding the desired equality (Theorem 1)". For completeness, Appendix A (in the paragraph between Lemma 4 and its proof) and B.2 respectively explain more precisely why we need such restrictions on F and that convergence is assumed without theoretical guarantee.

As suggested, we have clarified the position of our study with respect to standard RNN approaches in the BPTT background subsection, l. 95 and with Appendix A completed accordingly: "As detailed in Appendix A, the stability of the steady state implies that the gradients $\nabla_\theta^{\text{BPTT}}(t)$ decay (i.e. vanish) exponentially fast, which ensures that $\frac{\partial \mathcal{L}}{\partial \theta} = \sum_{t=0}^{T-1} \nabla_\theta^{\text{BPTT}}(t)$ converges, even if $T \to \infty$. In the context of convergent RNNs with a static input, vanishing gradients of BPTT are consequently not a problem and need not be addressed with LSTMs, as it is the case when learning from temporal data with RNNs".

**Reviewer 4**. The case of non symmetric weights is very interesting. It has been investigated in a real-time setting by Scellier et al. in their generalization of EP to vector-field dynamics (2018). As they show, learning still works in this setting, with forward and backward weights aligning throughout learning. We expect the same results in a discrete-time setting, with accelerated simulations, as we now have added it in the conclusion: "our discrete-time setting could also potentially accelerate simulations where the weights are not tied [Scellier, 2018]".

With regards to sequence tasks, we have added the following to the conclusion, l.219: "These results highlight that, in principle, EP can reach the same performance as BPTT on benchmark tasks - for RNN models *with fixed input*. One limitation of our theory however is that it has yet to be adapted to sequential data: such an extension requires to capture and learn correlations between successive equilibrium states corresponding to different inputs".

As suggested, we have added titles and legends to clarify each experimental set-up presented. For the sake of clarity, we have also specified the weight update implemented in the experiments, l. 169: "We study this architecture (...) with corresponding weight updates (14) and (16)". Moreover, we have also clarified when the effective weight update is being performed, l.113: "so that the effective weight update is performed at the end of the second phase." To improve the theory section, as also recommended by Reviewer 3, we have now provided a verbal outline of the proof of Theorem 1 - see answer to Reviewer 3. Finally, due to time and computational constraints inherent to hyperparameter tuning, we have not yet investigated what is the deepest network our algorithm can train, but we are addressing this now and aim to include these results in the camera-ready version.

[Meta-Review · NeurIPS 2019]

The authors first introduce a discrete-time version of equilibrium propagation (EP). They show the equivalence of EP with backpropagation through time (BPTT). They also apply it to a CNN (first time). They show step-by-step equality under certain conditions. All reviewers agree that the results are original, the quality and clarity of the paper is high, and the results are very significant for the NeurIPS community, in particular to researchers interested in biologically plausible replacements of backpropagation. This paper is a clear accept.